# *TUBG1* missense variants underlying cortical malformations disrupt neuronal locomotion and microtubule dynamics but not neurogenesis

Ekaterina L. Ivanova[1,2,3,4], Johan G. Gilet[1,2,3,4], Vadym Sulimenko[5], Arnaud Duchon [1,2,3,4], Gabrielle Rudolf[1,2,3,4], Karen Runge[1,2,3,4], Stephan C. Collins[1,2,3,4,6], Laure Asselin[1,2,3,4], Loic Broix[1,2,3,4], Nathalie Drouot[1,2,3,4], Peggy Tilly[1,2,3,4], Patrick Nusbaum[7], Alexandre Vincent[1,2,4], William Magnant[1,2,4], Valerie Skory[1,2,3,4], Marie-Christine Birling[8], Guillaume Pavlovic [8], Juliette D. Godin[1,2,3,4], Binnaz Yalcin [1,2,3,4], Yann Hérault [1,2,3,4], Pavel Dráber[5], Jamel Chelly [1,2,3,4,9,10,11] & Maria-Victoria Hinckelmann [1,2,3,4,11]

De novo heterozygous missense variants in the γ-tubulin gene *TUBG1* have been linked to human malformations of cortical development associated with intellectual disability and epilepsy. Here, we investigated through in-utero electroporation and in-vivo studies, how four of these variants affect cortical development. We show that *TUBG1* mutants affect neuronal positioning, disrupting the locomotion of new-born neurons but without affecting progenitors' proliferation. We further demonstrate that pathogenic *TUBG1* variants are linked to reduced microtubule dynamics but without major structural nor functional centrosome defects in subject-derived fibroblasts. Additionally, we developed a knock-in $Tubg1^{Y92C/+}$ mouse model and assessed consequences of the mutation. Although centrosomal positioning in bipolar neurons is correct, they fail to initiate locomotion. Furthermore, $Tubg1^{Y92C/+}$ animals show neuroanatomical and behavioral defects and increased epileptic cortical activity. We show that $Tubg1^{Y92C/+}$ mice partially mimic the human phenotype and therefore represent a relevant model for further investigations of the physiopathology of cortical malformations.

---

[1] Institut de Génétique et de Biologie Moléculaire et Cellulaire, 67400 Illkirch, France. [2] Centre National de la Recherche Scientifique, UMR7104, 67400 Illkirch, France. [3] Institut National de la Santé et de la Recherche Médicale, U1258, 67400 Illkirch, France. [4] Université de Strasbourg, 67000 Strasbourg, France. [5] Institute of Molecular Genetics of the Czech Academy of Sciences, Vídeňská 1083, 142 20 Prague, Czech Republic. [6] Université de Bourgogne, SVTE, Boulevard Gabriel, 21000 Dijon, France. [7] Service de Biochimie et de Génétique Moléculaire, Hôpital Cochin, AP-HP, Paris 75014, France. [8] CELPHEDIA, PHENOMIN, Institut Clinique de la Souris (ICS), 1 rue Laurent Fries, F-67404 Illkirch-Graffenstaden, France. [9] Fédération de Médecine Translationnelle, Université de Strasbourg, 67000 Strasbourg, France. [10] Laboratoire de Diagnostic Génétique, Hôpitaux Universitaires de Strasbourg, 67000 Strasbourg, France. [11] These authors jointly supervised this work: Jamel Chelly, Maria-Victoria Hinckelmann. Correspondence and requests for materials should be addressed to M.-V.H. (email: hinckelm@igbmc.fr)

The proper formation of the six-layered structure of the mammalian cortex depends on three highly regulated processes: proliferation of progenitor cells, migration of post-mitotic neurons and their maturation. Two main proliferative zones generate the majority of cortical neurons: the dorsal ventricular zone, where projection neurons are born, and the medial and caudal ganglionic eminences, where interneurons are generated. Projection neurons will then migrate radially along radial glial fibers[1], while interneurons will migrate tangentially over longer distances and independently of radial glial cells[2]. Alterations in these finely coordinated processes can lead to neurodevelopmental disorders including malformations of cortical development (MCDs), recognized as a frequent cause of epilepsy and intellectual delay[3].

Over the past decades, genetic studies have identified numerous causative mutations in subjects with MCDs and have greatly advanced our knowledge about the mechanisms and biological pathways involved in cortical formation and the pathophysiological mechanisms of MCDs[4,5]. These studies have outlined the importance of microtubules in every step of cortical development. Mutations in tubulins (TUBA1A, TUBB2B, TUBB3) and microtubule-associated proteins (LIS1, DCX, KIF2A, KIF5C, DYNC1H1) have been linked to a large spectrum of cortical malformations[6,7]. Moreover, microtubule organizing centers (MTOCs) such as the centrosome are highly implicated in the proper cerebral development, since multiple MCD-causing variants have been identified in proteins localizing at this organelle. Pathogenic variants in the pericentriolar and centrosomal proteins CDK5RAP2, CENPJ, CEP135, WDR62, CEP152, ASPM, and STIL have been discovered in subjects with primary microcephaly[8–13].

As far as centrosomal proteins are concerned, we have previously reported three missense variants in the γ-tubulin gene TUBG1, in subjects with posterior predominant pachygyria, corpus callosum abnormalities and microcephaly for two of the patients, associated with epilepsy and developmental delay[14]. A recent study reports four additional heterozygous pathogenic TUBG1 variants in subjects with similar clinical features and posterior cortical malformations[15]. In mammalian cells, γ-tubulin is highly conserved and often encoded by two genes. For instance, human γ-tubulin 1 and γ-tubulin 2 (TUBG1 and TUBG2) show, respectively, 98.9 and 97.6% amino acid sequence identity with the corresponding mouse isoforms[16]. TUBG1 is thought to be ubiquitously expressed while expression of TUBG2 appears to be restricted to the brain[16,17]. Both isoforms are concentrated at the centrosome[18–20]. Gamma-tubulin is a component of two characterized complexes γ-TuCs: the γ-tubulin small complex (γTuSC) and the γ-tubulin ring complex (γTuRC)[21]. γTuSC consists of a γ-tubulin dimer associated with one molecule of each GCP2 and GCP3. Multiple γTuSCs associate with other proteins (GCP4, GCP5, and GCP6) to form γTuRCs, which are recruited to MTOCs where they serve as template for microtubule nucleation[22–24]. Both γ-tubulin 1 and γ-tubulin 2 are found in γ-TuCs[20] and reportedly shown to nucleate microtubules[19]. However, the degree of their functional redundancy remains to be elucidated.

Here, we investigated the consequences of four human MCDs-related TUBG1 variants (Tyr92Cys, Ser259Leu, Thr331Pro, and Leu387Pro) on cortical development by using in utero electroporation and a knock-in Tubg1$^{Y92C/+}$ mouse model. We show that pathogenic TUBG1 variants affect neuronal positioning by disrupting neuronal migration without a major effect on progenitor proliferation. Our results suggest that disease-related TUBG1 variants exert their pathogenicity by affecting microtubule dynamics rather than centrosomal positioning or nucleation ability. Additionally, we report cortical and hippocampal neuroanatomical anomalies in our Tubg1$^{Y92C/+}$ animals, associated with behavioral alterations and susceptibility to epilepsy.

## Results

### TUBG1 pathogenic variants alter neuronal positioning.
Subjects with TUBG1 pathogenic variants present with abnormal cortical thickness and layering, suggesting neuronal mispositioning arising from alterations during cortical development. To investigate the effect of pathogenic TUBG1 variants on neuronal positioning in vivo, we used in utero electroporation to induce overexpression of the four pathogenic variants (Fig. 1a) under the control of a CAG promoter, together with a GFP-encoding reporter in progenitors. We confirmed co-expression of the fluorescent reporter with all TUBG1 pathogenic variants and their stable overexpression, by immunohistochemistry in electroporated neurons and immunoblot in Neuro2a cells (Supplementary Fig. 1a–d). Mice cortices were electroporated at E14.5 and the distribution of electroporated cells was analyzed after 4 days. In brain sections expressing the control-empty vector, the majority of the electroporated neurons were positioned within the upper layers of the cortical plate (CP). Overexpression of human WT-TUBG1 showed a normal pattern of distribution. On the contrary, cells expressing any disease-related variant were mainly localized in the intermediate zone (IZ), with almost no electroporated cells reaching the CP (Fig. 1b, c). Arrested neurons express Cux1, an upper cortical layer marker, while they are negative for deep layer markers (CTIP2, TBR1) (Fig. 1d), suggesting they are committed to upper-layers cortical projection neurons. Default in positioning persists at both early (P8) and relatively mature postnatal stages (P20). Similarly to our data at E18.5, abnormally localized cells in the white matter express only upper-layer markers (Fig. 1e and Supplementary Fig. 1e). This indicates that the phenotype is not a transient phenomenon and is probably due to an arrest rather than a delay in neuronal migration.

Given the similarity between TUBG1 and TUBG2, we characterized their expression profile during cortical development. Tubg1 mRNA levels in mouse cortices remain relatively constant throughout embryonic development, whereas TUBG2 mRNA levels are low during early embryonic stages and progressively increase starting from E16.5 (Supplementary Fig. 2a). Additionally, we investigated a possible functional redundancy of the two isoforms during cortical development. First, we tested whether TUBG1 and TUBG2 can recover the phenotype shown to be associated to TUBG1 knock-down[14] by co-electroporating sh-Tubg1 with either TUBG1, TUBG2 or pathogenic TUBG1. Overexpression of TUBG1 or TUBG2 alone did not induce a particular phenotype in neuronal positioning (Supplementary Fig. 2b). Inversely, downregulation of Tubg1 leads to misplacement of neurons, which at E18.5 are mainly localized in the IZ. This phenotype was significantly recovered when WT-TUBG1 was expressed but barely modified by TUBG2 co-expression, and even accentuated by co-electroporation of TUBG1-Tyr92Cys (Supplementary Fig. 2c). We then performed a series of complementation experiments by co-electroporating the Tyr92Cys TUBG1 variant with increasing concentrations of either TUBG1 or TUBG2. Co-electroporation of either isoforms leads to a rescue in the neuronal positioning defect, nonetheless, while a significant restauration of the phenotype was obtained with TUBG1, rescue with TUBG2 was only partial and dose-dependent (Supplementary Fig. 2d). However, TUBG2 rescue was significant for other mutants (Supplementary Fig. 2e). Altogether, these results suggest limited functional overlap between the two isoforms during cortical development.

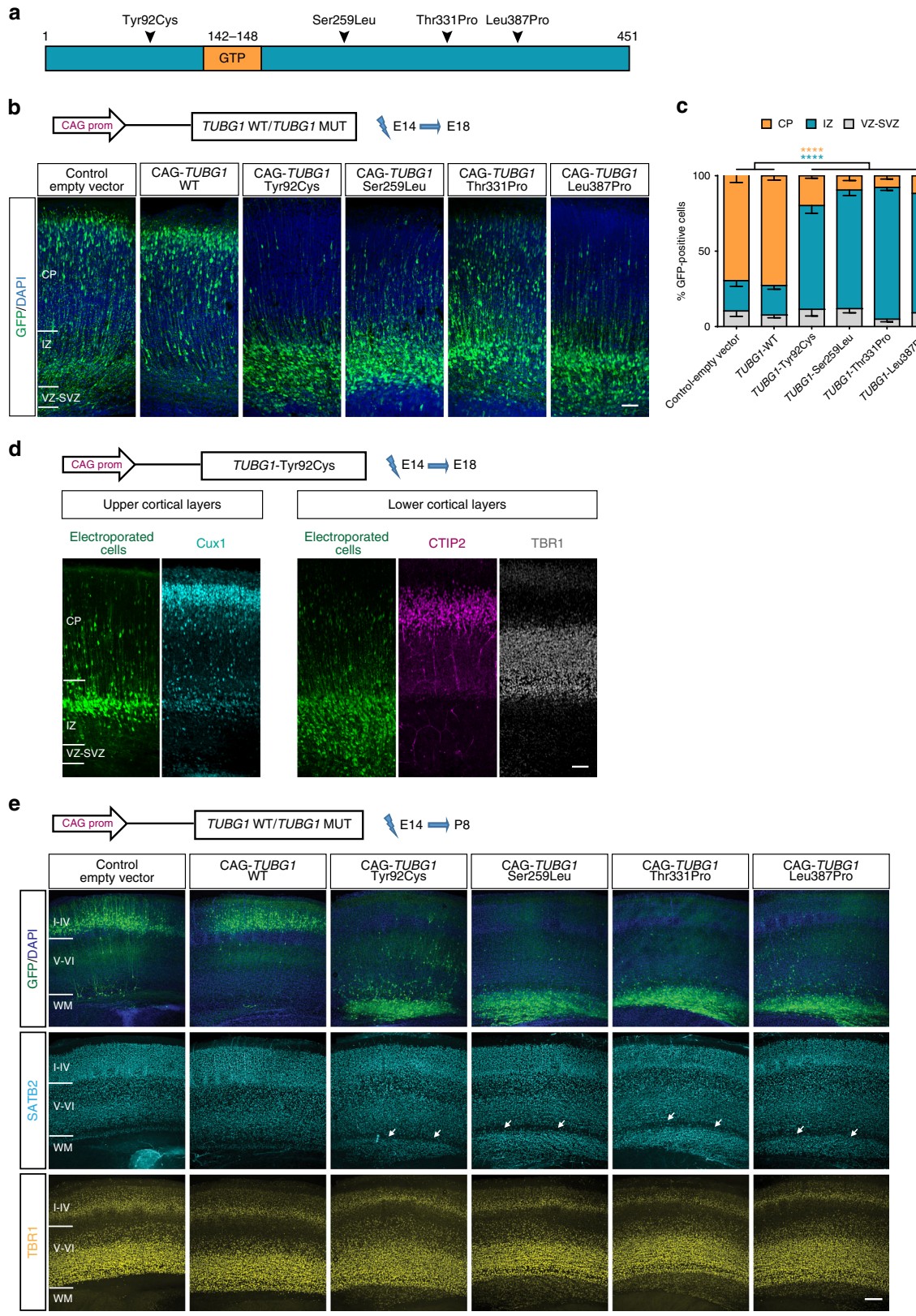

**Post-mitotic processes underlie abnormal positioning**. To better understand mechanisms underlying the observed positioning defects, we studied the effects on progenitor cells and on post-mitotic neurons. We first analyzed whether *TUBG1* pathogenic variants affect proliferation by evaluating the percentage of mitotic progenitors. For three of the variants, no difference in mitotic index was observed. However, the Thr331Pro variant increased in the number of VZ progenitors in mitosis (Supplementary Fig. 3a). To determine whether this is due to an increased number of cells entering mitosis or a decreased number of cells exiting the cell cycle, we performed a cell cycle exit assay (Supplementary Fig. 3b), focusing on two of the variants,

**Fig. 1** Pathogenic variants in *TUBG1* alter neuronal positioning. **a** Linear representation of TUBG1 polypeptide. Black arrows indicate relative position of the four investigated mutations. **b** Coronal sections of E18.5 brains electroporated at E14.5 with either a control-empty vector, the WT form or one of the four *TUBG1* mutated forms and co-electroporated with a GFP-encoding reporter (green), stained with DAPI (blue) Scale bar 50 μm. **c** Percentage of fluorescent cells in three different regions: CP, IZ, VZ/SVZ. Data from at least three embryos per condition were analyzed with Two-way ANOVA with Tukey's multiple comparisons test, ****$p < 0.0001$, compared to control-empty vector or TUBG1 wild type overexpression. **d** Coronal sections of E18.5 brains electroporated at E14.5 with the Tyr92Cys variant and stained for the upper-layer marker Cux1 (cyan), deep layers markers CTIP2 (magenta), TBR1 (gray) and DAPI (blue). Scale bar 50 μm. **e** Coronal sections of P8 brains electroporated at E14.5 with either a control-empty vector or one of the four mutant *TUBG1* forms. Sections were stained for the upper-layer marker SATB2 (cyan), lower layer marker TBR1 (yellow) and with DAPI (blue). White arrows show ectopic SATB2$^+$ cells in the white matter (WM). Scale bar 250 μm. CP, cortical plate; IZ, intermediate zone; VZ/SVZ, ventricular zone/subventricular zone. Data are represented as mean ± s.e.m. Source data are provided in the Source Data file

Tyr92Cys presenting no proliferation alteration and Thr331Pro, with an increased mitotic index. We observed that the Thr331Pro variant induced a decrease in the number of progenitors exiting the cell cycle with a tendency of increasing the progenitors having undergone S-phase (Edu+), suggesting a prolonged cell cycle, while no significant alterations were observed for the Tyr92Cys variant (Supplementary Fig. 3b–e). Also, we performed Pax6/Tbr2 immunolabelling and observed no alteration in the percentage of apical (Pax6$^+$/Tbr2$^-$) and intermediate (Pax6$^-$/Tbr2$^+$) progenitors, nor in the percentage of generated neurons (Pax6$^-$/Tbr2$^-$) (Supplementary Fig. 3f), suggesting that *TUBG1* variants do not affect the balance between proliferative and neurogenic divisions of progenitor cells. Finally, we analyzed the structure of radial glial fibers and confirmed for all conditions, they are correctly formed and attached to the pial surface (Supplementary Fig. 3g). Altogether, these data indicate no major effect of three out of the four studied variants on progenitor cells.

In view of these results, we studied the pathogenicity of *TUBG1* variants specifically in post-mitotic neurons. We electroporated WT or *TUBG1* human pathogenic variants under the control of the weak neuron-specific promoter Doublecortin (DCX)[25] to avoid overexpression (Supplementary Fig. 4a), with a strong NeuroD-GFP reporter, and studied neuronal positioning at E18.5. Brain sections expressing either control-empty vector or *TUBG1*-WT showed normal distribution of electroporated cells (Fig. 2a, b). However, for pathogenic variants, a large proportion of electroporated cells was accumulated in the IZ, and this accumulation was less severe for Thr331Pro variant. Interestingly, for all mutant forms, many electroporated cells were abnormally located in the lower layers of the CP (Fig. 2c). We also confirmed persistence of the phenotype at early (P3 and P8) and at relatively mature postnatal stages (P21). Yet again, mispositioned neurons express upper rather than deep cortical layer markers (Supplementary Fig. 4c–e). Together, these data support the idea of a disrupted neuronal migration, possibly due to defects in post-mitotic processes.

We also assessed neuronal positioning in the hippocampus by in utero electroporation of the ammonic neuroepithelium at E14.5. Neurons electroporated with WT-*TUBG1* were found exclusively within the pyramidal layer at P8 (Fig. 2d). Neurons electroporated with *TUBG1*-Tyr92Cys were however partially located within the stratum oriens. NeuN and CTIP2 staining revealed they were differentiated into mature pyramidal neurons (Fig. 2d and Supplementary Fig. 4f). Thus, *TUBG1* variants affect migration of pyramidal neurons both in the neocortex and the hippocampus.

**Disease-causing variants affect neuronal migration dynamics.** To further understand disrupted neuronal positioning, we studied neuronal migration using time-lapse videomicroscopy to follow the dynamics in morphology and locomotion of migrating neurons exiting the ventricular zone. We focused our studies on the

Tyr92Cys and the Thr331Pro variants. For both, we observed a strikingly reduced number of neurons in the locomotion phase compared to control-empty vector and WT-*TUBG1*, with the majority of neurons remaining immobilized at the edge of the IZ (Fig. 3a, b and Supplementary movies 1–4). We measured the speed and pausing for migrating neurons and observed no major differences for average locomotion velocity compared to WT (Fig. 3c). Nonetheless, expression of the Tyr92Cys variant increased pausing and time spend in pause (Fig. 3d, e). We observed that the majority of arrested neurons presented a bipolar morphology which was maintained throughout the 10 h of recording. However, we did not detect any defects in morphology transition preceding the migration process (Fig. 3f, g).

Given the importance of TUBG1 in the organization of the centrosome and the key role of this organelle for neuronal migration, we examined centrosome dynamics in both migrating and non-migrating neurons in mouse cortices by co-electroporation with the PACT-mKO1 construct allowing labelling of the centrosome (Fig. 3h). No differences in centrosome positioning were observed for migrating neurons between the control and pathogenic *TUBG1*. Most of these migrating neurons display the centrosome ahead of the nucleus within the leading process followed by a forward displacement of the soma (Fig. 3i). On the other hand, centrosome localization in non-migrating neurons differs between control and *TUBG1*-Tyr92Cys conditions. While control neurons' centrosome is mainly localized randomly or behind the nucleus, for the Tyr92Cys variant there was an abrupt increase in cells with centrosome located ahead of the nucleus (from 10.4% in the control to 42.8% in Tyr92Cys condition) (Fig. 3j). Further analysis showed that these cells are mainly bipolar. Indeed, this configuration (non-migrating bipolar neurons with centrosome ahead of the nucleus) was only observed in 5% of control neurons but in over 50% of the Tyr92Cys neurons (Fig. 3k). Together, these results highlight a population of immobilized neurons that although adopting the correct bipolar morphology and correctly positioning their centrosome, fail to initiate glial-guided locomotion.

**Disrupted locomotion linked to altered microtubule dynamics.** We performed series of in cellulo experiments aiming to investigate a potential effect of *TUBG1* pathogenic variants on microtubule organization and dynamics. First, we analyzed the subcellular localization of recombinant TUBG1 using a *TUBG1*-RFP fusion construct. As shown in Supplementary Fig. 5a, all four variants are concentrated at the centrosome of differentiated Neuro2a cells and are also diffusely found in the cytosol, similarly to the WT.

We then analyzed microtubule organization in fibroblasts derived from a skin biopsy from two of the subjects (bearing the Tyr92Cys and the Thr331Pro variants) and compared them with three different unrelated control fibroblasts. We observed normal microtubule network with microtubule arrays emerging from

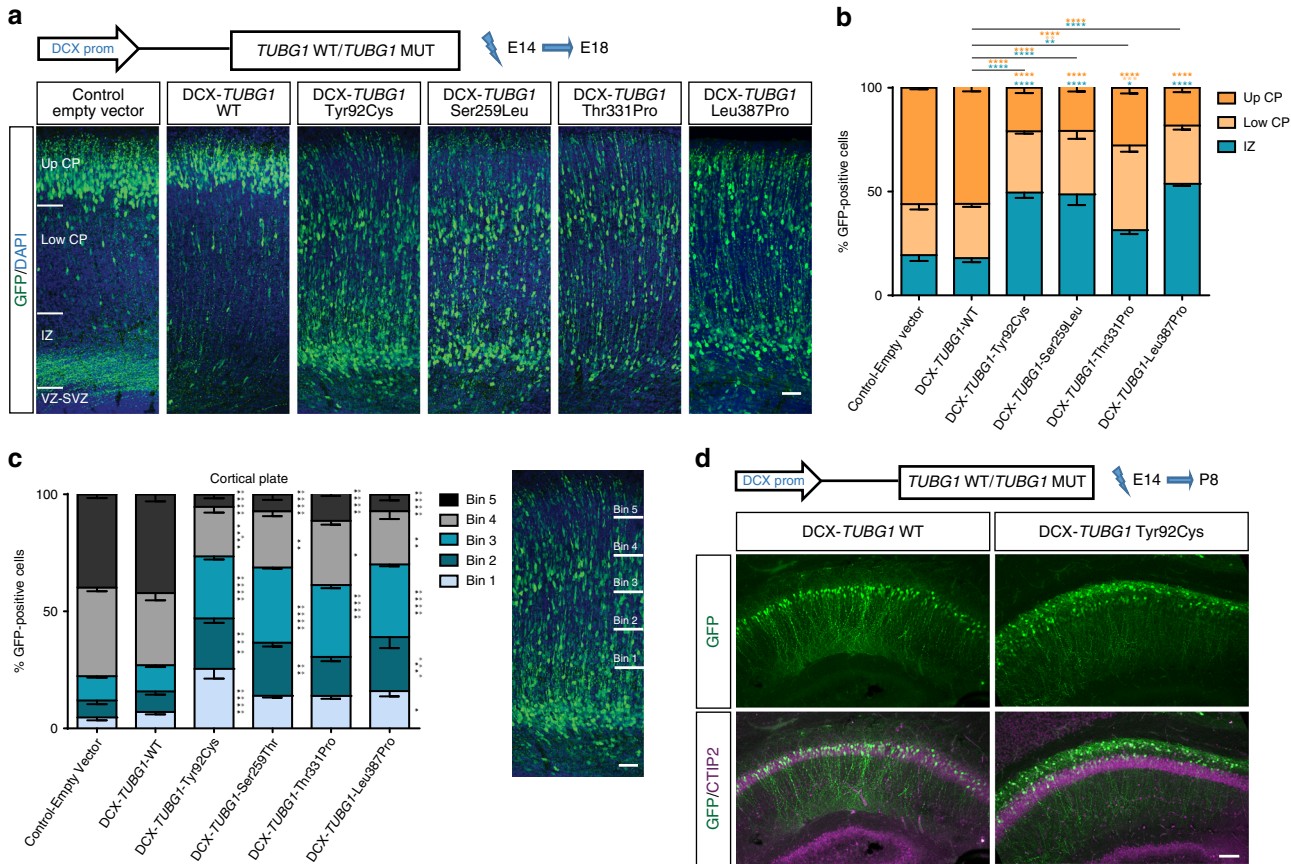

**Fig. 2** Neuron-specific expression of *TUBG1* mutations affects neuronal positioning **a** Coronal sections of E18.5 brains electroporated at E14.5 with either a control-empty vector, WT-*TUBG1* or one of the four mutated forms of *TUBG1* under the control of a neuron-specific DCX promoter and co-electroporated with a reporter construction encoding for GFP under the control of a neuron-specific NeuroD promoter (green). Sections were stained with DAPI (blue). CP, cortical plate; IZ, intermediate zone; VZ/SVZ, ventricular zone/subventricular zone. Scale bar 50 μm. **b** Quantification of fluorescent neurons in three different regions: up CP, down CP, and IZ for at least four electroporated embryos per group, Two-way ANOVA with Tukey's multiple comparisons test, *$p < 0.05$, **$p < 0.01$, ***$p < 0.001$, ****$p < 0.0001$, compared to control-empty vector or wild type TUBG1 overexpression. **c** Quantification of fluorescent neurons in the cortical plate. The image on the right shows the cortical plate divided in five equal sections (bins). Scale bar 50 μm. GFP-positive cells were counted in each bin for at least four embryos per condition, Two-way ANOVA with Tukey's multiple comparisons test, *$p < 0.05$, **$p < 0.01$, ***$p < 0.001$, ****$p < 0.0001$, black asterisks compared to control-empty vector and gray asterisks compared to TUBG1 wild type overexpression. **d** In utero electroporation of DCX-*TUBG1*-Y92C or WT together with GFP (green) in the ammonic neuroepithelium at E14.5. Images show coronal sections of the CA1 region of the hippocampus at P8 counterstained with CTIP2 marker (magenta). Scale bar 150 μm. Data are represented as mean ± s.e.m. Source data are provided in the Source Data file

centrosomes (Fig. 4a). Fibroblasts did not present any shape or morphology abnormalities. However, after induced depolymerization, we saw that newly nucleated microtubules were shorter and with less organized structure in affected subjects' fibroblasts, compared to longer and straight control microtubules (Fig. 4b, c). Furthermore, we used these cells to analyze the centrosomal structure and organization by electron microscopy. We observed no significant difference in patients' fibroblasts compared to control ones (Supplementary Fig. 5b).

We then investigated the effect of *TUBG1* variants on microtubule dynamics using videomicroscopy after transfecting plus end-binding protein EB3-GFP in HeLa cells[26]. Though overexpression of WT-*TUBG1* led to a slight decrease in microtubule dynamics, we observed a more drastic decrease in microtubule polymerization rate for the three variants Tyr92Cys, Ser259Leu and Thr331Pro (Fig. 4d, e and Supplementary movies 5–10) significantly different from both control-empty vector and wild type overexpression. Similar results were obtained with subject-derived fibroblasts bearing the Tyr92Cys and the Thr331Pro variants (Fig. 4f, g and Supplementary movies 11–15). These data show that *TUBG1* mutations decrease microtubule

dynamics which could pave the way to better explain the altered locomotion phenotype.

Finally, we investigated the capacity of mutant mouse Tubg1 (mTubg1) to form γ-tubulin complexes by performing immunoprecipitations with anti-TagRFP antibody from extracts of Neuro2A cells expressing either mTubg1-TagRFP (WT control) or its pathogenic variants. Western blots were immunostained for RFP, GCP2 (marker for γTuSC), GCP4 (marker for γTuRC) and total γ-tubulin. Results revealed quantitative differences in proteins co-precipitated with mTubg1-TagRFP. When compared with WT, lower amounts of GCP2, GCP4, and γ-tubulin were associated with mutated variants of mTubg1-TagRFP, while similar amounts of TagRFP-tagged γ-tubulins were detected in all samples (Fig. 4h, Supplementary Fig. 5c, d). Reciprocal immunoprecipitation with anti-GCP2 antibody confirmed reduced association of mutated variants with GCP2 (Fig. 4i). Similar amounts of co-precipitated GCP2, GCP4, and endogenous γ-tubulin were detected in all samples. Altogether, these results suggest that mutant recombinant TUBG1 has a decreased ability to dimerize and to form γ-tubulin complexes.

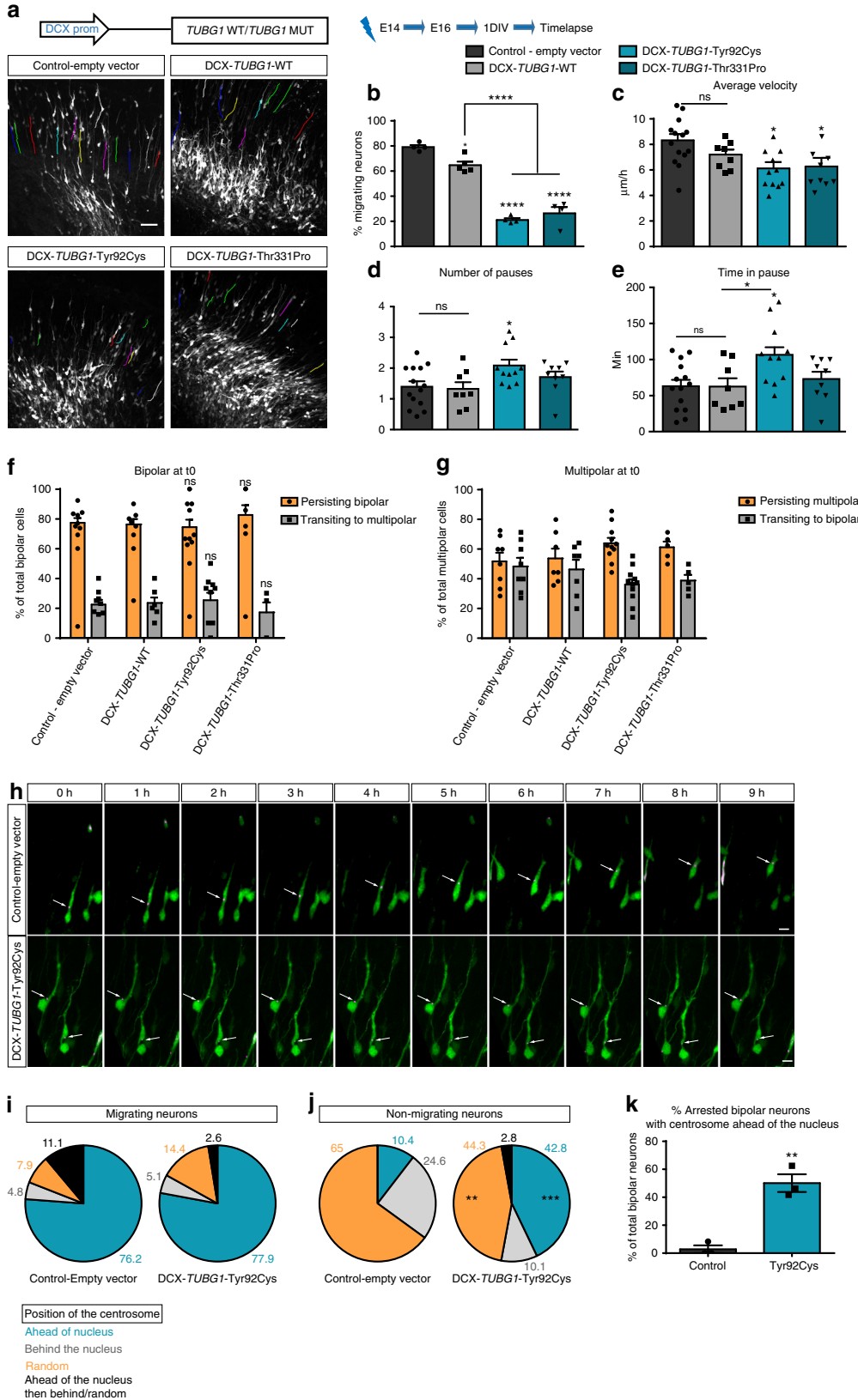

**Knock-in $Tubg1^{Y92C/+}$ embryos show abnormal cortical layering**. To further study the effects of $Tubg1$ variants in vivo, we generated a constitutive knock-in mouse model ($Tubg1^{Y92C/+}$) bearing the first identified disease-causing mutation Tyr92Cys. In this model, the Tyr92Cys variant is constitutively expressed at a

heterozygous state on the C57 black 6 (C57BL/6 N) genetic background (Supplementary Fig. 6a). To validate our model, we performed RT-PCR, and subsequent qPCR and Sanger sequencing, as well as immunoblots. We confirmed heterozygous presence of the mutation and comparable mRNA levels between

**Fig. 3** Real-time recordings of migrating cortical neurons. **a** Locomotory paths (colored lines) of electroporated neurons recorded during time-lapse imaging. Scale bar 50 μm. **b–e** Percentage of migrating neurons ($n = 4$ independent time-lapse experiments per group) (**b**), average migration velocity ($n \geq 8$ independent time-lapse experiments per group) (**c**), average number of pauses ($n \geq 8$ independent time-lapse experiments per group) (**d**), and total time spent in pause ($n \geq 8$ independent time-lapse experiments per group) (**e**) during the 10 h of recording. Data were analyzed with One-way ANOVA with Tukey's post hoc test, $*p < 0.05$, $**p < 0.01$, $****p < 0.0001$ compared to control-empty vector; $****p < 0.0001$ compared to DCX-*TUBG1*-WT. **f** Percentage of electroporated bipolar cells persisting bipolar or transitioning to multipolar morphology in 10 h recordings ($n = 4$ independent experiments per group). Comparisons were made to control-empty vector and TUBG1 wild type overexpression. **g** Percentage of electroporated multipolar cells persisting multipolar or transitioning to bipolar morphology in 10 h recordings ($n = 5$ independent experiments per group). Comparisons were made to control-empty vector and TUBG1 wild type overexpression, Two-way ANOVA with Tukey's multiple comparisons test. **h–k** Centrosome positioning in migrating and non-migrating neurons. **h** Sequences of images during time-lapse recording of bipolar neurons co-electroporated with GFP (green) and the centrosomal marker PACT-mKO1 (magenta) together with either control or *TUBG1*-Tyr92Cys. Centrosome is indicated by white arrows. Scale bar 10 μm. Quantification of centrosome positioning in migrating (**i**) and non-migrating (**j**) neurons. Neurons were divided in four different categories according the position of their centrosome during the time-lapse recording: ahead of the nucleus, behind the nucleus, random relative to the nucleus, ahead of the nucleus in the beginning of the recording and then behind it or random. Data are represented as pie charts for either migrating or non-migrating neurons. Two-way ANOVA with Sidak's multiple comparisons test, $**p < 0.01$, $***p < 0.001$. **k** Percentage of arrested electroporated bipolar neurons with the centrosome ahead of nucleus, $n \geq 3$ independent time-lapse experiments per group, Unpaired $t$-test, $p = 0.0024$, compared to control-empty vector. Histograms represent mean ± s.e.m. Source data are provided in the Source Data file

mutant and WT allele, as well as stability of the protein (Supplementary Fig. 6b–e). $Tubg1^{Y92C/+}$ mice were born in the expected Mendelian ratios and developed normally with a usual lifespan. Interestingly, both male and female mice exhibited sub-fertility.

Since in utero electroporation results showed a major effect of disease-causing *TUBG1* variants on neuronal migration, we first investigated whether $Tubg1^{Y92C/+}$ mice display layering anomalies at embryonic stages. We analyzed thickness of the CP and different layers by immunohistochemistry on E18.5 brains (Fig. 5a). While we observed no difference in total CP thickness between the WT and $Tubg1^{Y92C/+}$ mice, a more detailed analysis revealed an increase in layer V thickness with a concomitant reduction of layer VI in $Tubg1^{Y92C/+}$ mice. Additionally, CTIP2 staining showed that $Tubg1^{Y92C/+}$ mice present with visibly abnormal organization of layer V, which is larger and with lower cell density, while it presents with a more compact organization in control mice. On the other hand, we also observed a small population of heterotopic SATB2$^+$ cells at the interphase of the IZ and the CP of mutant embryos (Fig. 5b). Given this observation, we assessed neuronal migration by labelling a cohort of progenitors with EdU at E14.5 and examining their localization four days after in the SVZ/VZ, the IZ and in the six layers of the CP that we divided in five equal bins (Fig. 5c). We observed a higher percentage of EdU-positive cells in the IZ at the limit with layer VI with a corresponding depletion of cells in bins 3, 4, and 5 of the CP (Fig. 5d). Arrested cells appeared as a visible band in section images and were positive for SATB2 (Fig. 5e), consistent with our previous observations.

We then analyzed cell proliferation in WT and mutant embryonic brains. We found no significant differences in progenitors' proliferation in any of the studied developmental stages (E12, E14, E16) (Supplementary Fig. 7a). Quantification of the pool of apical and intermediate progenitors as well as newborn neurons in the VZ/SVZ at E14 and E16 revealed no differences between the wild type and knock-in mice (Supplementary Fig. 7b, c).

Finally, we studied the dynamics of radial migration by time-lapse recordings in WT and $Tubg1^{Y92C/+}$ organotypic brain cultures. We observed that the pathogenic variant leads to a significant reduction in the percentage of migrating neurons exiting the IZ (Fig. 5f, g and Supplementary movies 16–17). Moreover, while there were no defects in the velocity of migration, neurons from $Tubg1^{Y92C/+}$ mice paused more often and for longer periods of time (Fig. 5h–j). We could not observe any defects in neuronal polarity transition before migration (Fig. 5k, l). Arrested

neurons mainly exhibit the bipolar morphology required to initiate radial migration, with a significantly increased number of them containing the centrosome ahead of the nucleus (Fig. 5m). In line with our in utero electroporation data, together these results suggest that cortical layering defects related to TUBG1 mutations result from altered neuronal migration. Neurons bearing Tyr92Cys mutation adopt the correct morphology and centrosome localization, but appear unable to initiate locomotion.

**$Tubg1^{Y92C/+}$ mice exhibit neuroanatomical defects.** Using a recently developed robust approach for the assessment of 166 brain parameters across 22 distinct brain regions[27] (Supplementary Dataset 1), we analyzed neuroanatomy in 16-weeks-old male $Tubg1^{Y92C/+}$ mice by systematically quantifying the same sagittal brain region at Lateral + 0.60 mm. Overall size anomalies were identified in mutant mice with a reduction in size of all assessed parameters when compared to WT littermates (Fig. 6a). Total brain area was significantly reduced by 14.8%, concomitantly with smaller cortices (−14.1%, Fig. 6b). Importantly, area and thickness of the secondary motor cortex decreased by 10 and 9%, respectively (Fig. 6c). Additional assessment of the thickness of cortical layers revealed a significant decrease of 23% in the width of layer I and a decrease of 20% in layer VI. By contrast, thickness of layer V increased by 11% but did not reach the level of statistical significance ($P = 0.1$). Noteworthy, we were unable to delineate the boundary between layers II–III and V in two knock-in mice, possibly due to abnormal laminar disorganization. White matter structures were also decreased and included for example the fimbria of the hippocampus (−18%, $P = 0.0052$) and the anterior commissure (−26%, $P = 0.017$). The cerebellum was one of the most severely affected structures exhibiting a decreased size of 27% (Fig. 6b). Interestingly, our analysis additionally showed an abnormal organization of the hippocampal CA1 region with presence of heterotopic neurons in the stratum oriens region positioned above the pyramidal layer. Positive staining for the neuronal marker NeuN, suggested they are differentiated hippocampal pyramidal neurons (Fig. 6d). No defects in interneurons number nor distribution were found in $Tubg1^{Y92C/+}$ mice (Supplementary Fig. 8).

The time scale of $Tubg1^{Y92C/+}$ mouse microcephaly was further investigated through a similar morphological screen at postnatal day 0 (P0). Relying on the adult brain screen, we performed morphometric analysis for 11 distinct brain regions (Supplementary Dataset 2). However, we were unable to detect a global microcephaly, and observed no visible heterotopia in the

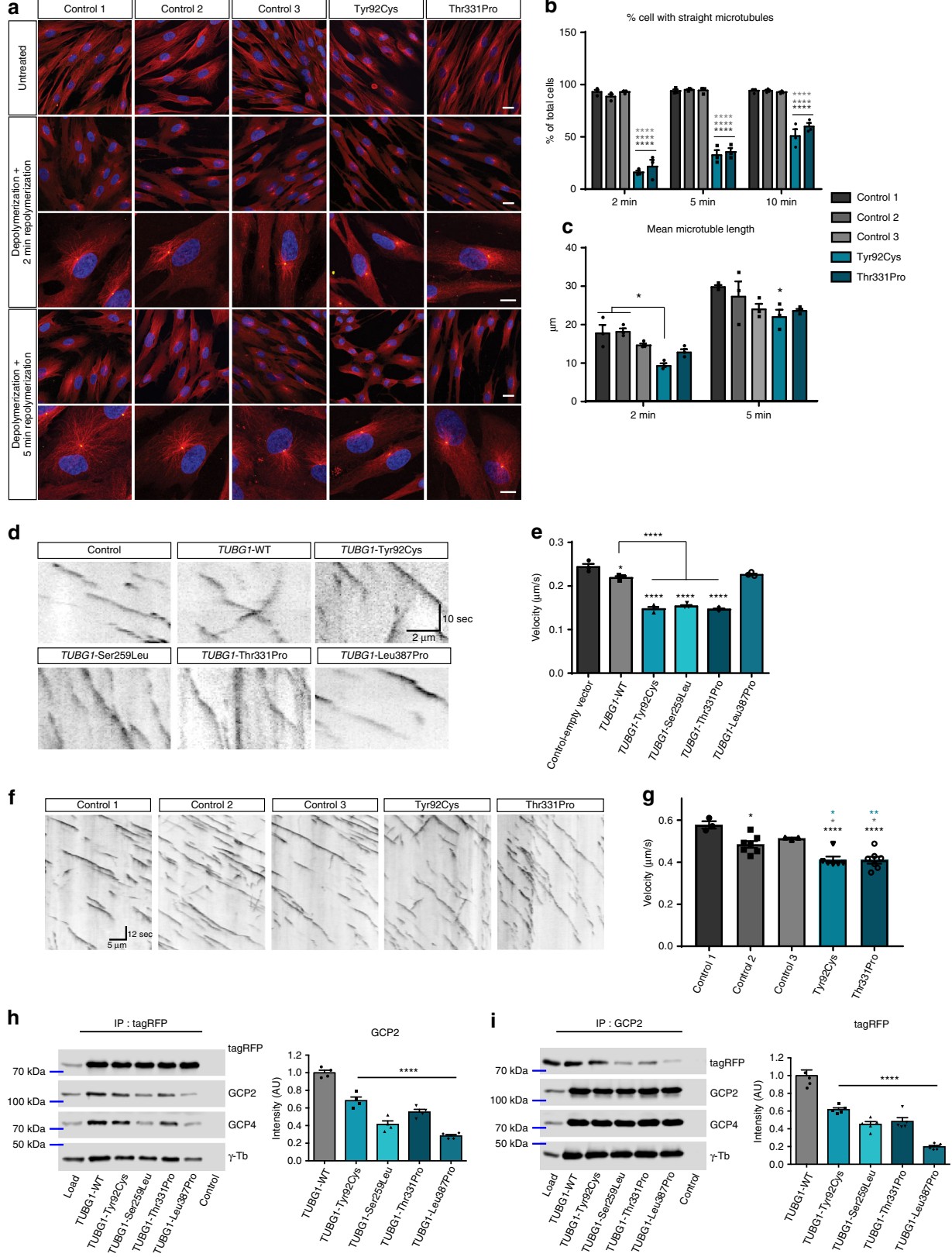

hippocampus (Supplementary Fig. 9a). Further analysis showed that this hippocampal anomaly only becomes apparent at early post-natal stages but not before birth of pups (Supplementary Fig. 9b) consistent with the previously described slower, post-natal neuronal migration in the hippocampus[28].

**$Tubg1^{Y92C/+}$ mice show behavioral and epileptic anomalies.** Given the neuroanatomical abnormalities observed in the $Tubg1^{Y92C/+}$ mice, and the fact that cognitive function is impaired in subjects with MCDs, we tested whether this correlates with behavioral abnormalities by performing a battery of tests allowing

**Fig. 4** *TUBG1* mutations alter microtubule dynamics and γ-tubulin complexes **a** Microtubules (red) and TUBG1 (yellow) staining in subject-derived fibroblasts, without treatment and after microtubule depolymerization and subsequent repolymerization. Scale bars 20 μm (upper panels) and 10 μm (lower panels). **b**, **c** Quantification of the percentage of cells presenting with straight and organized microtubules (**b**) and microtubule mean length (**c**), Two-way ANOVA with Tukey's multiple comparisons test, *$p < 0.05$, **** $< 0.0001$, compared to controls. **d**, **e** Microtubule dynamics in HeLa cells transfected with EB3 and different *TUBG1* constructs. **d** Kymographs show EB3 trajectories in time (y, sec) and space (x, μm). Scale bar $Y = 10$ s, $X = 2$ μm. **e** Histograms represent mean velocity of EB3 comets, $n = 3$ independent experiments, one-way ANOVA with Tukey's multiple comparisons test, *$p < 0.05$, ****$p < 0.0001$ compared to control-empty vector or wild type overexpression. **f**, **g** Microtubule dynamics in subject-derived fibroblasts. **f** Representative Kymographs of dynamic EB3 comets in different controls and mutant fibroblasts. Kymographs are calibrated in time (y, sec) and space (x, μm). Scale bar $Y = 12$ s, $X = 10$ μm. **g** Quantification of EB3 mean velocity, $n \geq 3$ independent experiments, one-way ANOVA with Dennett's post hoc test, *$p < 0.05$ compared to control fibroblasts. **h**, **i**. Immunoprecipitations in Neuro2a cells expressing tagRFP-tagged mouse *Tubg1* (WT control) or its mutated variants. **h** Left: representative immunoblot after immunoprecipitation with anti-tagRFP antibody. Right: densitometric quantification of immunoblots stained against GCP2. Relative intensities of samples with mutated Tubg1-TagRFP normalized to WT control and to tagRFP levels in individual samples. **i** Left: representative immunoblot for immunoprecipitation with anti-GCP2 antibody. Right: densitometric quantification of immunoblots stained against RFP. Relative intensities of samples with mutated Tubg1-TagRFP normalized to WT control and to GCP2 levels in individual samples. Controls in left panels contained proteins A without antibodies incubated with the cell extracts, $n = 5$ independent experiments, one-way ANOVA with Bonferroni's multiple comparisons test, ****$p < 0.0001$. All data are presented as mean ± s.e.m. Source data are provided in the Source Data file

to evaluate locomotor activity, repetitive behaviors, learning and memory and epileptic susceptibility. We observed a hyper-locomotion for *Tubg1*[Y92C/+] mice compared to WT littermates, with an increase in both travelled distance and rearing, in the repetitive behaviours (Fig. 7a, b) and in the circadian activity paradigms (Supplementary Fig. 10a–h). This finding was supported by an increased distance travelled in the open field (OF) assay (Supplementary Fig. 10i–k). Additionally, we performed a novel object recognition (NOR) task, based on the innate exploratory behaviour of mice[29], to assess their long-term memory capacity. *Tubg1*[Y92C/+] mice presented significantly impaired NOR performance, spent more time sniffing the two familiar objects (FO) during the training session and their recognition index (RI) was around zero, consistent with lack of preference for the novel object (NO) (Fig. 7c, d). Next, to challenge hippocampus mediated cognitive behaviours, we tested mice in a fear conditioning (FC) assay, exploring their associative memory[30]. When tested for either contextual or cued fear, *Tubg1*[Y92C/+] mice displayed significantly reduced immobility time suggesting impaired fear expression (Fig. 7e). No phenotype was observed for digging and grooming behaviors (Supplementary Fig. 10l, m), nor when we assessed short-term memory of KI mice in the Y-maze paradigm (Supplementary Fig. 10n, o).

Finally, we assessed epileptic activity in *Tubg1*[Y92C/+] animals. Superficial cortical EEG recordings were visually inspected to detect epileptiform discharge events. Spontaneous cortical spike-waves were observed in all *Tubg1*[Y92C/+] mice but not in WT controls. No spontaneous seizure was recorded in both strains. We therefore challenged animals with the seizure-provoking agent PTZ to determine their susceptibility to seizure. PTZ, an inhibitor of the γ-aminobutyric acid type 1 (GABAA) receptor, is the most widely used model of acute chemical induced seizures[31]. We administrated intraperitoneally a pro-convulsive dose of 30 mg kg$^{-1}$ and recorded EEG during 30 min after drug administration. One minute after injection, generalized tonico-clonic seizure was recorded in all analyzed *Tubg1*[Y92C/+] but not in WT mice (Fig. 7f).

Overall, our behavioral analysis highlight reduced neurodevelopmental performances illustrated by an increased locomotor and rearing activities with impaired object recognition and associative learning. These results suggest that *Tubg1*[Y92C/+] mice exhibit common features of mouse models of intellectual disability. In addition, *Tubg1*[Y92C/+] mice showed abnormal epileptic activity. Altogether these findings are in agreement with the observed cortical and hippocampal abnormal structures and propose that *Tubg1*[Y92C/+] mice represent a relevant model that partially mimics human phenotype of MCDs.

## Discussion

Gamma-tubulin is a major component of the centrosome in mammalian cells[32,33]. In mammals, two genes code for γ-tubulin —*TUBG1* and *TUBG2*, both localizing at centrosomes and nucleation competent[19]. Identification of *TUBG1* pathogenic variants in subjects with cortical malformations[14,34] led us to study the underlying mechanisms by performing a combination of in utero and in vivo functional studies. All 7 MCD-related identified variants in *TUBG1* are heterozygous missense variants, occurring de novo, suggesting that they exert their pathogenicity through a dominant mechanism, rather than through a loss of function. Up to date no truncating variants have been identified in *TUBG1*. This is in contrast with the other described variants in centrosomal proteins in subjects with primary microcephaly which are majorly at a homozygous state[35]. Furthermore, homozygous *Tubg1* knock-out mice present defects in mitosis during early embryogenesis and die at the morula/blastocyst stages, while heterozygous *Tubg1*[+/−] mice develop normally and present with no histological defects[16]. In agreement with this, our studies of *Tubg1*[+/−] mice showed no major neuroanatomical nor behavioral abnormalities (Supplementary Figs 6f, g and 11). Finally, our data from in utero electroporation suggest that WT-TUBG1 is able to partially rescue the mutation-induced defect, suggesting a possible dominant-negative mechanism of the variants. Altogether, these observations suggest that haploinsufficiency is not associated with neurodevelopmental pathologies in mice and therefore a knock-out model is not representative of disease-causing missense variants in *TUBG1*.

We have developed a knock-in *Tubg1* mouse model with the aim to better mimic the human MCD phenotype. Our *Tubg1*[Y92C/+] mice present with altered cortical layering during embryonic stages in agreement with our in utero electroporation data showing an accumulation in the IZ of E14.5 born neurons expressing *TUBG1* pathogenic variants. At adult stage *Tubg1*[Y92C/+] mice present with a global microcephaly, abnormal cortical layering and defects in hippocampal organization. Correlating with these observations, *Tubg1*[Y92C/+] mice exhibit memory deficits in the NOR and the FC paradigms.

It is worth mentioning that several mouse models of MCDs related to abnormal neuronal migration present with anomalies in the hippocampal structure[36–38]. For example, heterozygous *Tubb2b* mice with a point variant, present with disorganized CA3 area, associated with cognitive defects and long-term potentiation deficits[39]. Similarly, the CA3 region of hemizygous *Dcx* mice showed heterotopic neurons in both the stratum radiatum and the stratum oriens with milder abnormalities in the CA2 and the CA1 regions[36]. Interestingly, neither of these mice are reported to

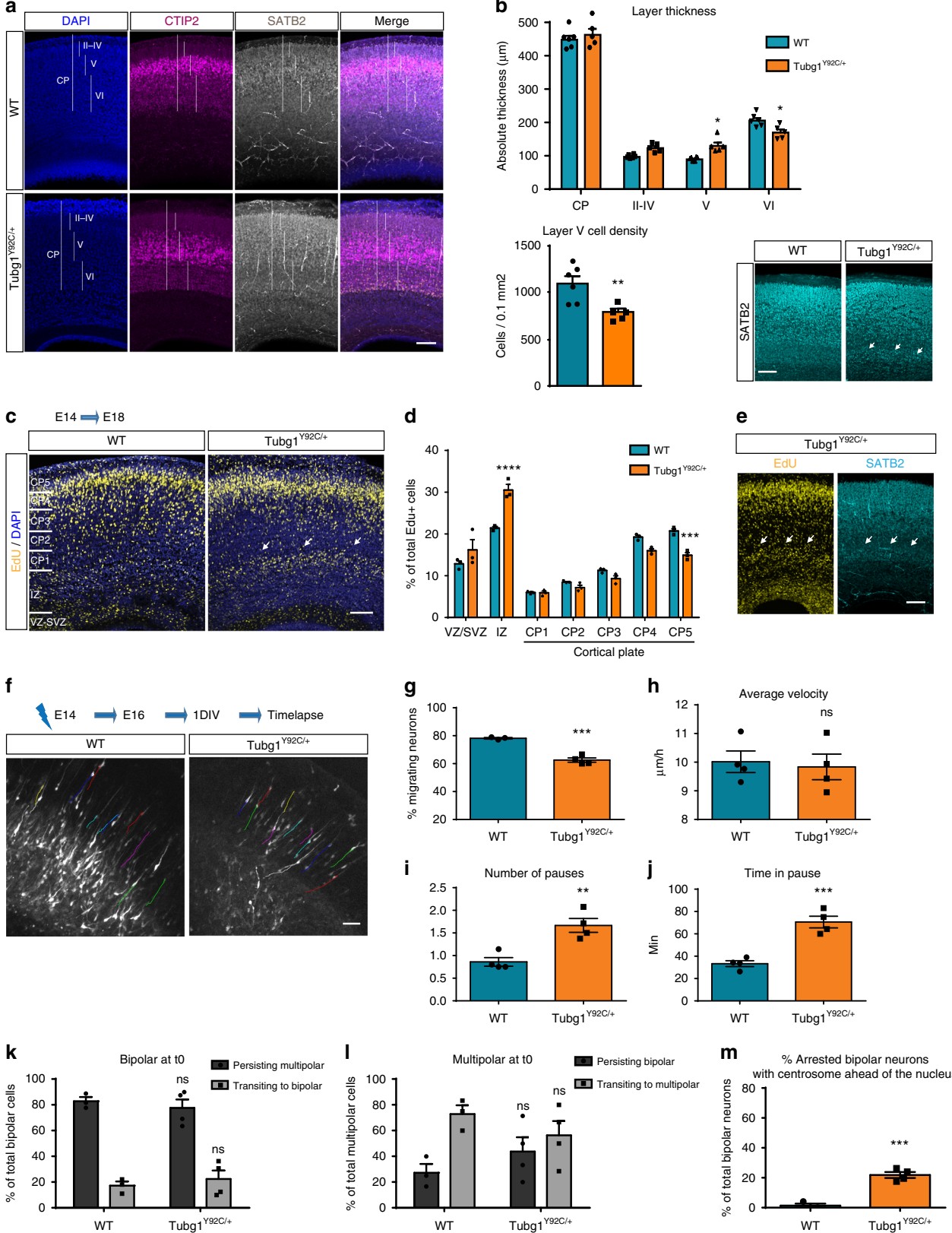

present major abnormalities in the neocortical architecture and lamination, even though these genes have been involved in neuronal migration in the neocortex using in utero electroporation[40,41]. Additionally, very few of MCD mouse models have been reported to have pathological epileptic seizures, although they do show cognitive deficits[42]. Our *Tubg1*[Y92C/+]

mice exhibit increased susceptibility to PTZ-induced epileptic seizures originating in the neocortex and could therefore represent a relevant model to further investigate pathophysiological mechanisms underlying MCD-related epilepsy. It remains to be investigated whether they are due to an abnormal connectivity of the heterotopic pyramidal neurons. The ability of heterotopic

**Fig. 5** $Tubg1^{Y92C/+}$ mice show abnormal cortical layering during development. **a** E18.5 cortices stained against SATB2 (gray), CTIP2 (magenta) and DAPI (blue). Vertical lines depict how cortical layers were considered for quantification, scale bar 50 μm. **b** Histograms represent the thickness of the cortical layers and CP (upper panel) and the cell density of layer V (lower panel). Data are from at least five embryos per condition, unpaired *t*-test, *$p < 0.05$, **$p < 0.01$. Right image shows heterotopic SATB2$^+$ cells. Scale bar 50 μm. **c** EdU staining in the cortex. Cortical plate was divided into five equal bins from the intermediate zone to the marginal zone. White arrows show EdU-positives cells accumulation in the IZ. CP, cortical plate; IZ, intermediate zone; VZ/SVZ, ventricular zone/subventricular zone. Scale bar 100 μm. **d** Quantification of EdU-positive cells in the specified regions, $n = 3$ embryos per group, Two-way ANOVA with Tukey's multiple comparisons test, *$p < 0.05$, ***$p < 0.001$. **e** Representative image of SATB2 staining in cortices of EdU injected $Tubg1^{Y92C/+}$ mice. Scale bar 50 μm. **f** Locomotory paths (colored lines) of electroporated neurons during time-lapse imaging. Scale bar 50 μm. **g** Percentage migrating neurons, $n = 3$ WT and $n = 4$ $Tubg1^{Y92C/+}$; **h** average migration velocity, **i** average number of pauses, and **j**. total time spent in pause, during the 10 h of recording, $n = 4$ independent experiments per group. Unpaired *t*-test, **$p < 0.01$, ***$p < 0.001$ compared to wild type mice. **k** Percentage electroporated bipolar cells persisting bipolar or transitioning to multipolar morphology in 10 h recordings ($n = 3$ WT, $n = 4$ $Tubg1^{Y92C/+}$ independent experiments per group). **l** Percentage electroporated multipolar cells persisting multipolar or transitioning to bipolar morphology in 10 h recordings ($n = 3$ WT, $n = 4$ $Tubg1^{Y92C/+}$, independent experiments per group). **m** Percentage of arrested electroporated bipolar neurons with the centrosome ahead of nucleus, $n = 3$ WT, $n = 4$ $Tubg1^{Y92C/+}$ independent experiments per group. Unpaired *t*-test, ***$p < 0.001$ compared to wild type mice. All data are presented as mean ± s.e.m. Source data are provided in the Source Data file

neurons to form one hand their classic afferent and efferent connections, and on the other, additional local connections has been discussed by Chevassus-au-Louis and Represa (1999)[43]. Authors point out that this could lead to dual connectivity patterns between regions that are not normally connected, inducing functional alterations such as epilepsy. Abnormal post-natal connectivity could additionally contribute to the global microcephaly, since this microcephaly appears postnatally in $Tubg1^{Y92C/+}$ mice. In line with this, for MCD patients with TUBG1 mutations, for whom HC size at birth was available, values appeared within the normal range (p.L387P, HC = 31 cm; p.Y92C, HC = 33 cm), but then progressed to severe reduction (p.L387P, HC = −5,5 SD; p.Y92C, HC = −4SD), suggesting a secondary microcephaly that mainly develops after birth. It would therefore be relevant to investigate whether the neuroanatomical abnormalities are due to additional defects in postnatal processes with the development of a conditional KI model allowing timed-regulated expression of $Tubg1$-Tyr92Cys. Overall, our $Tubg1^{Y92C/+}$ model presents with cortical and hippocampal malformations associated with a global microcephaly, behavioral alterations and epileptic susceptibility, all features that correlate with the phenotype of affected subjects making it an interesting model for gaining insight into the mechanisms of MCDs.

Gamma-tubulin is essential for the formation of mitotic spindle microtubules[44–48] but also for the coordination of mitotic events such as metaphase, anaphase and cytokinesis[49], so we expected to observe proliferation defects related to TUBG1 pathogenic variants. The centrosome is known to be highly implicated in cell division where it plays a major role in organizing mitotic microtubules and variants in centrosomal components could be expected to affect primarily progenitor proliferation. Indeed, several mouse models with partial or complete loss of function of centrosomal proteins have been characterized with neurogenic defects and subsequent cell death leading to microcephaly, due to centrosome and mitotic spindle abnormalities[50,51]. Surprisingly, in our in utero electroporation experiments we saw that three of the TUBG1 variants did not compromise proliferation of progenitor cells. Similarly, there were no disruptions in progenitor proliferation in $Tubg1^{Y92C/+}$ mice. These observations raise the question of whether the identified TUBG1 mutations affect the overall structure or interactions within the centrosomal machinery. Interestingly, no structural defects were found in centrosomes and no subcellular mislocalization was observed for any of the four recombinant forms. Moreover, our videomicroscopy studies showed correct centrosome positioning in both migrating and non-migrating bipolar neurons, suggesting that the pathogenic phenotype is probably not related to the centrosomal function of TUBG1.

Additionally, our EB3 studies showed reduced microtubule dynamics for three of the recombinant forms and a slight decrease that did not reach statistical significance for the Leu387Pro variant, probably due to a lack of sensitivity of the technique. Moreover, after complete depolymerization, newly nucleated microtubules were disorganized and shorter in human fibroblasts bearing the TUBG1 variants. However, since no major abnormalities were observed in the overall organization of interphase microtubules without depolymerization, we hypothesize that microtubule nucleation properties of the centrosome are not affected and the observed deficiencies are rather due to altered microtubule dynamics. Immunoprecipitation assays showed that pathogenic forms co-precipitate with fewer γTuSC and γTuRC components, as well as with fewer γ-tubulin, suggesting a reduced capacity for mutant recombinant γ-tubulin to dimerize and to integrate these complexes. In addition to these observations, given the dominant mechanism of the variants, we hypothesize that mutant γ-tubulin affects the role of γ-TuCs and probably their interactions in regulating microtubule dynamics causing the pathogenic phenotype. Centrosome structure and γ-tubulin nucleating capacity do not seem to be affected by TUBG1 variants, which could explain why mitotic processes and cell divisions are preserved. Our data suggests that disrupted neuronal migration could be related to additional non-centrosomal functions of γ-tubulin. In line with this hypothesis, centrosome-independent roles for γ-tubulin in the control of microtubule dynamics has been reported in *D. melanogaster*[52]. Authors show that γTuRCs localize along interphase microtubules and suggest they stabilize them by limiting catastrophe events at the plus tips. Similarly, in *S. pombe*, γ-tubulin has been suggested to cooperate with the kinesin-like protein *Pkl1p* to regulate the dynamics and organization of microtubules[53]. In HeLa cells, specific inhibition of γ-tubulin with gatastatin decreased the dynamics of interphase microtubules without affecting the overall microtubule network ant the number of growing microtubules[54]. In post-mitotic neurons, γTuRCs have been shown to participate in centrosome-independent microtubule nucleation mediated by Augmin complexes[55]. Finally, during cortical development, it was recently shown that acentrosomal microtubules nucleation controls neuronal morphology ensuring proper migration in vivo[56]. Altogether, this supports our results showing that pathogenic gamma-tubulin leads to reduced microtubule dynamics during interphase, and disrupts neuronal migration.

## Methods

**DNA constructs.** Commercially available Human untagged TUBG1 cDNA (NM_001070.3) cloned in pCMV6-XL5 vector (SC119462) was purchased from Origene. Variants (p.Tyr92Cys, p.Ser259Leu, p.Thr331Pro and p.Leu387Pro)

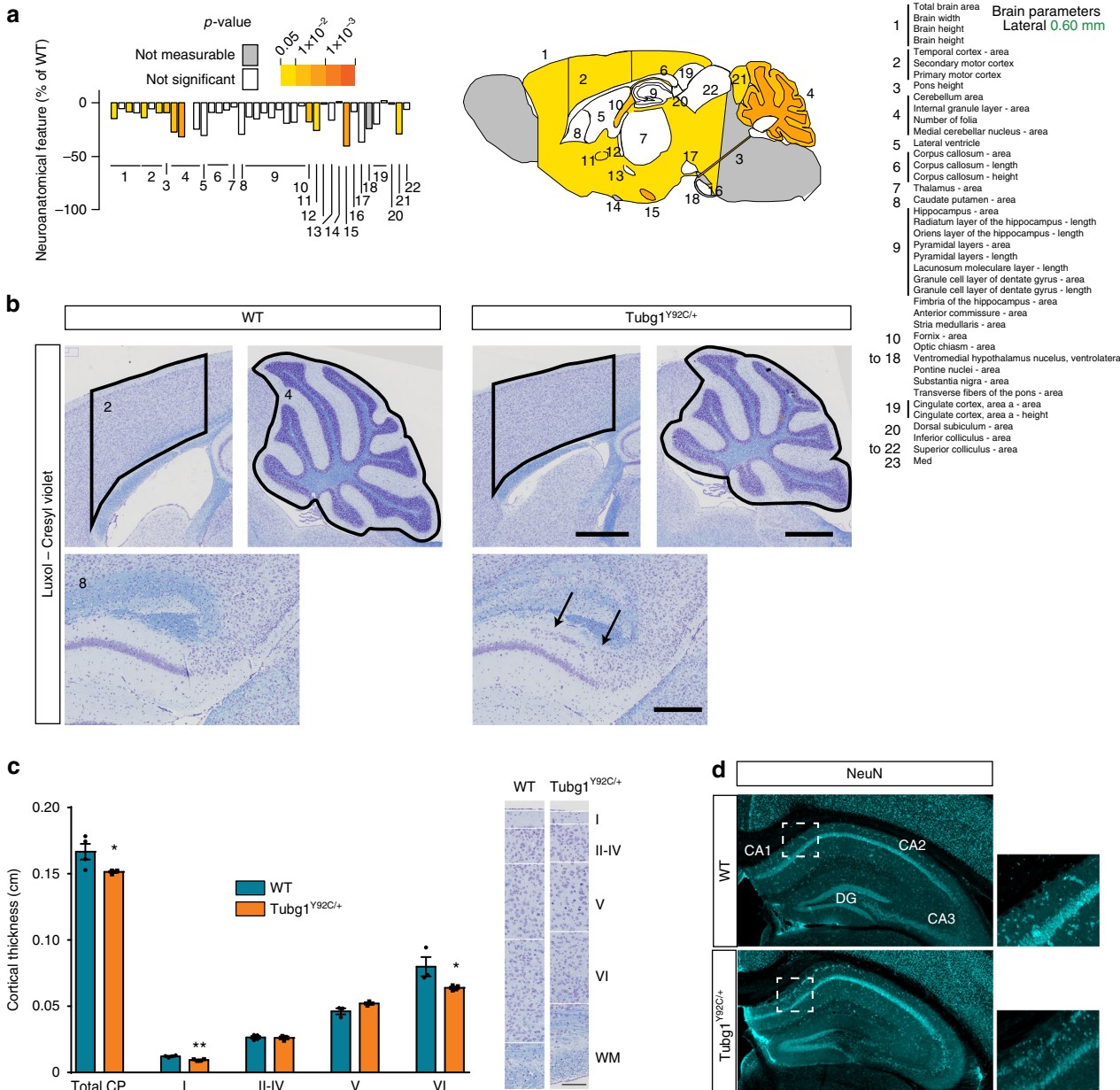

**Fig. 6** Microcephaly and abnormal hippocampal organization in *Tubg1*[Y92C/+] mice. **a** Schematic representation of affected brain regions plotted in sagittal planes according to *p*-values. The right image represents a section at Lateral + 0.60 mm. White coloring indicates a *p*-value higher than 0.05 and gray shows not enough data to calculate a *p*-value. Histograms are showing the percentage decrease (minus scale) in size of measured brain regions as compared to the controls (100%). Numbers indicate assessed brain regions listed on the right; *n* = 4 16-weeks-old male mice per group. **b** Sagittal sections of the hippocampus, the cortex and the cerebellum from control WT and *Tubg1*[Y92C/+] adult mice stained with cresyl violet/luxol blue. Black arrows indicate the hippocampal heterotopia. Scale bar 0.1 cm (upper panels), 200 μm (lower panel). **c** Cortical layers analysis (left panel) performed in sections of the secondary motor cortex in control WT and Tubg1[Y92C/+] adult mice stained with cresy violet/luxol blue, *n* = 4 mice per group, unpaired *t*-test, **p* < 0.05, ***p* < 0.01. Data are presented as mean ± s.e.m. Source data are provided in the Source Data file. Representative images showing layers defects (right panel). Scale bar 150 μm. **d** Coronal sections of the hippocampus from control WT and *Tubg1*[Y92C/+] adult mice stained with NeuN, scale bar 300 μm. Higher magnification view of the region delimited with dotted square is shown on the right of each image, scale bar 100 μm

were introduced by site-directed mutagenesis using QuikChange Site-Directed Mutagenesis Kit (Agilent Technologies). WT and mutated cDNAs encoding human *TUBG1* were then inserted in the multiple cloning site of psiSTRIKE vector under the control of a CAG promoter. The DCX promoter from a DCX-IRES-GFP vector, kindly provided by U. Mueller (The Scripps Research Institute, CA, USA), was cut with SpeI and EcoRI and inserted into the psiSTRIKE vector, replacing the CAG promoter, resulting in a construct encoding the WT and mutated human *TUBG1* under a DCX promoter. Vector pmTubg1-TagRFP encoding mouse γ-tubulin 1 fused to Tag-RFP was prepared as described. Shortly, coding sequences of mouse Tubg1 were cut out from pmTubg1-FLAG vector by *Eco*RI/*Sal*I restriction enzymes and ligated into pCI-TagRFP[19]

resulting in vector encoding TagRFP-tagged mouse γ-tubulin 1 (pmTubg1-TagRFP). Construct was verified by sequencing. Lentiviral vector pmTubg1-TagRF with puromycin resistance, encoding mouse γ-tubulin 1 fused to TagRFP was prepared as described[57]. Shortly, cassette encoding mouse γ-tubulin fused to TagRFP was digested out from pmTubg1-TagRFP construct[19] (see above) by *Eco*RI/*Not*I and ligated to pCDH-CMV-MCS-EF1-puro vector (System Biosciences) resulting in the lentiviral construct pmTubg1-TagRFP-puro. BLBP-GFP plasmid was kindly provided by N. Heintz (The Rockefeller University, NY, USA). The following plasmids were used as reporters for IUE experiments: a pCAGGS-IRES-Tomato (RFP) vector; a pCIG2-IRES-GFP vector, kindly provided by R.Belvindrah (INSERM, Paris, France); a NeuroD-GFP vector, kindly

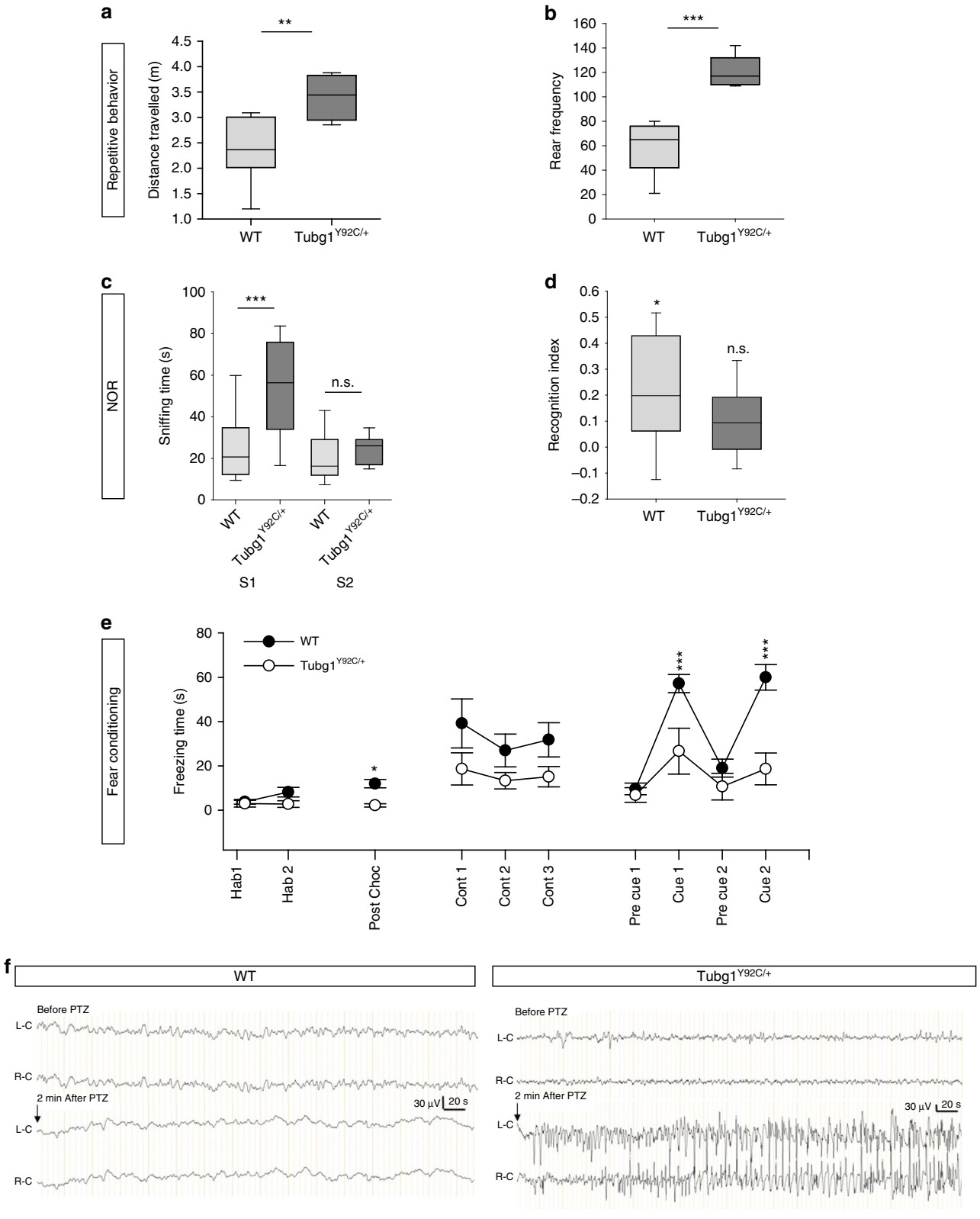

provided by J. Godin (IGBMC, Strasbourg, France), a pSilencer-CAG-Venus-2 and a pCAG-mScarlet vectors, kindly provided by Julien Courchet (Institut NeuroMyoGene, Lyon). pCAG-PAKT-mKO bearing the pericentrin-AKAp450 centrosomal targeting (PACT) domain fused to Kusabira Orange was kindly provided by J. Godin (IGBMC, Strasbourg, France). All plasmid DNAs were prepared using the EndoFree plasmid purification kit (Macherey Nagel). Silencing experiments were performed using a sh-RNA sequence targeting mouse *Tubg1* mRNA (5′-TGAACCTGATAGCCACAGACCGCCTACAC-3′) that was cloned in the psiStrike vector under the control of a U6 promoter.

**Antibodies**. The following antibodies were used in this study: γ-tubulin (aa 38–53) GTU-88 (T6557, mouse, used at 1/10000 for WB and 1/1000 for IF/IHC, Sigma); Gamma-tubulin (434–451) TU-30 (ab27074, mouse, used at 1/1000, Abcam); GAPDH (MAB374, mouse, used at 1/1000, Chemicon); Cux1 (sc-13024, rabbit, used at 1/100, Santa Cruz); Cux1 (11733-AP, used at 1/100, Proteintech); GFP (A10262, chicken, used at 1/1000, ThermoFisher); CTIP2 (ab18465, rat, used at 1/500, Abcam); NeuN (MAB377, mouse, used at 1/500, Millipore); SATB2 (ab51502, mouse, used at 1/400, Abcam); TBR1 (ab31940, rabbit, used at 1/500, Abcam); Pax6 (PRB-278P, rabbit, used at 1/100, Eurogentec); Tbr2 (14–4875–80,

**Fig. 7** $Tubg1^{Y92C/+}$ mice present with behavioral abnormalities **a, b** Repetitive behaviours. Distance travelled (**a**) and occurrences of rearing (**b**) numbered during 10 min of observation in a novel home-cage environment showing a significant increase for $Tubg1^{Y92C/+}$ mice. **c, d** Novel object recognition (NOR) task. Mice were allowed to explore an identical pair of objects, and after 24 h, they were presented with the familiar object and a new object. The sniffing time in seconds during the 14 min of the presentation and test session are represented in box plots. (TIT 24 h, nose close to the object <1 cm, L) (**c**). The recognition index (RI) (**d**) was calculated as the percentage of time sniffing time novel object (NO) minus the time sniffing the familiar object (FO) divided by the total time of sniffing. In box plots center line represents the median, end of boxes 25th and 75th percentile and whiskers 10th and 90th percentiles. **e** Fear conditioning test. Graphs plot the freezing during the different sessions of the test. The 4 min of habituation (Hab1 and 2) and the 2 min post choc. The 6-min context session (Cont 1, 2, and 3) was run 24-h after conditioning. The 8-min cue session was performed 5-hours after the context session. A sequence of 2-min with no cue (pre cue1 and 2) and 2 min with light/auditory (cue 1 and 2) conditioning stimulus were repeated twice. Data are presented as mean ± s.e.m., $n = 9$ mice per genotype. One-way ANOVA (**a**, **b**, and **d**) or Two-way ANOVA (**c**, **e**, and **f**) *$p < 0.05$, **$p < 0.01$, ***$p < 0.001$. Source data are provided in the Source Data file. **f.** Representative EEG trace. Left panel: EEG recording before and after PTZ in WT mice without discharge activity. Right panel: $Tubg1^{Y92C/+}$ mice epileptiform discharge events before PTZ and generalized tonicoclonic seizure after PTZ. L-C: left cortex; R-C: right cortex. $n = 3$ adult mice from each genotype

rat, used at 1/200, eBioscience); PH3 (06–570, rabbit, used at 1/500, Millipore); Ki67 (NCL-L-Ki67-MM1, mouse, used at 1/500, Leica); Anti-tRFP (AB234, rabbit used at 1/5000 for WB, Evrogen); GCP4 (sc-271876, mouse, used at 1/1000 for WB, Santa Cruz); Anti-mouse antibody conjugated with HRP (W402B, goat, used at 1/10000 for WB, Promega); Anti-rabbit antibody conjugated with HRP (W401B, goat, used at 1/100000 for WB, Promega). Anti-GCP2 antibody GCP2-01 (mouse monoclonal IgG2b) used for immunoprecipitation was described previously[58]; Anti-GCP2 antibody GCP2-02 (mouse monoclonal IgG1, in the form of hybridoma spent culture supernatant, used at 1/10 for WB) was described previously[58].

**Cell culture**. Mouse Neuroblastoma N2a and HeLa cells provided by the cell culture platform of the IGBMC (Strasbourg) are mycoplasma free (PCR test Venorgem) and have not been authenticated. N2a cells and subject's human fibroblasts cells were cultured in DMEM (Gibco) supplemented with 1 g L$^{-1}$ of glucose, 5% foetal calf serum and gentamycine (DMEM-1 g L$^{-1}$ glucose-5% FCS-gentamycine). Transfection was performed using Lipofectamine 2000 (Invitrogen) according to the manufacturer's instructions and cells were then cultured for 48 h. N2a cells were used for differentiation assays 24 h after transfection by replacing the medium with DMED without serum and supplemented with 1 µM Retinoic acid. After 48 h cells were fixed in 4% PFA. For lentivirus production, HEK 293FT packaging cells (Invitrogen) were grown at 37 °C in 5% CO2 in DMEM supplemented with 10% FCS and antibiotics. The cells used for lentivirus production were at passage 4–15. For depolymerization assays, human fibroblasts were incubated with 5 µM nocodazole in DMEM for 1 h, at 4 °C. Medium was then replaced with preheated DMEM-1 g L$^{-1}$ glucose-5% FCS-gentamycine. Cells were then fixed at different time points: 0, 5, and 5 min.

**Lentiviral infection**. Lentiviral infections were done as described previously[59]. Briefly, aliquots of 1.4 ml Opti-MEM medium (Invitrogen) were mixed with 21 µl of ViraPower Lentiviral Packaging Mix (Invitrogen), 14 µg lentiviral construct and 82 µl of Lipofectamine 2000. The mixture was incubated for 20 min at room temperature before it was added to semi-confluent HEK-293FT packaging cells in 150 cm$^2$ tissue-culture flask. After 3 days, viruses in culture supernatants were concentrated by centrifugation at 25,000 rpm for 2 h using JA-25.50 rotor (Beckman Coulter, Palo Alto, CA). The pellet was resuspended in 1 ml of culture medium and 0.2 ml aliquot was added to 7 ml of medium in 25 cm$^2$ tisue-culture flask with cells. The transfection mixture was replaced after 3 days with fresh complete medium containing 5 µg ml$^{-1}$ puromycin. Stable selection was achieved by culturing cells for 1–2 weeks in the presence of puromycin.

**Immunoprecipitation**. To prepare extracts for immunoprecipitation, cells were rinsed twice in cold HEPES buffer (50 mM HEPES [pH 7.6], 75 mM NaCl, 1 mM MgCl2, and 1 mM EGTA) and extracted for 10 min at 4 °C with HEPES buffer supplemented with 1% Nonidet P-40 (extraction buffer), protease inhibitor mixture (Complete EDTA-free, Roche) and phosphatase inhibitors (1 mM Na$_3$VO$_4$, 1 mM NaF). The suspension was spun down (20000 × $g$, 15 min, and 4 °C) and supernatant was collected. Protein quantification in samples was assessed with a bicinchoninic acid (BCA) protein assay kit (Thermo Scientific). Samples were diluted with TBST to the final protein concentration 0.6 mg ml$^{-1}$. Immunoprecipitations were performed as described elsewhere[60]. Extracts were incubated with Protein A beads (GE-Health Care) saturated with (i) rabbit antibody to TagRFP; (ii) monoclonal antibody GCP2–01 to GCP2; or (iii) immobilized protein A alone. 7.5% SDS-PAGE and immunoblotting were performed as described previously[61]. Bound primary antibodies were detected after incubation of the blots with HRP-conjugated secondary antibodies. HRP signal was detected with chemiluminescent reagents (Thermo Scientific) and the LAS 3000 imaging system (Fujifilm). AIDA image analyzer (ver.4) software (Raytest) was used for quantification of signals from immunoblots.

**Western blot**. Cells were lysed in RIPA buffer (50 mM Tris-HCl, pH 7.7, 0.15 M NaCl, 1 mM EDTA and 1% Triton X-100) supplemented with proteases inhibitors (Roche). Mouse cortices were lysed in RIPA buffer supplemented with proteases inhibitors (Roche) using a syringe with an 18 G needle. Protein concentrations were determined with a Bradford assay. Equal amounts of lysates were then loaded on polyacrylamide gels and transferred onto nitrocellulose membranes. Membranes were blocked in 5% milk solution or 2% BSA in TBS-0.1% Tween and then incubated with the primary antibodies overnight at 4 °C. HRP (horseradish)-conjugated secondary antibodies (Thermo Fischer) were used at 1:10000. Uncropped and unprocessed scans of blots can be found in Supplementary Fig. 12.

**Electron microscopy**. Samples were immersed in glutaraldehyde (2.5%) and paraformaldehyde (2.5%) in cacodylate buffer (0.1 M, pH 7.4) and post-fixed in 1% osmium tetroxide, dehydrated through graded alcohol (50, 70, 90, and 100%) and propylene oxide for 30 min each, and embedded in Epon 812. Semithin sections were cut at 2 µm on an ultra microtome (Leica Ultracut UCT) and ultrathin sections were cut at 70 nm and contrasted with uranyl acetate and lead citrate and examined at 70 kv with a Morgagni 268D electron microscope (Phillips, FEI Electron Optics, Eindhoven, the Netherlands). Images were captured digitally by Mega View III camera (Soft Imaging System).

**EB3 tracking analyses**. Patient's fibroblasts and HeLa cells were cultured in fluorodish™ FD35-100 (World Precision Instruments) and transfected with EB3-GFP plasmid using Lipofectamine 2000 (Invitrogen) following manufacturer instructions. Live video-microscopy recording was performed using a Confocal Spinning disk—NikonTi—Roper iLas FRAP system equipped with a PLAN APO VC 60x objective (Nikon Instrument, Inc., Melville, USA) and driven by Metamorph 7.0 (Molecular Devices, LLC). Samples were maintained at 37 °C and 5% CO2 during acquisition. Images were collected with Photometrics Prime 95B™ Scientific CMOS Camera (PHOTOMETRICS) every 200 ms during 2 min. Movie assembly and time projections were done using ImageJ software (http://rsb.info.nih.gov/ij/, NIH). For generation and analyses of kymographs Image J plug-in kymotoolbox (Available on demand on http://www.bic.u-bordeaux.fr) was used. Kymographs were calibrated in time ($y$ in sec) and space ($x$ in µm) and analyses were performed manually by following EB3 comet trajectories. Mean speed was extracted by the plug-in.

**qPCR**. Total RNA was prepared in three independent isolations from the cortices of mouse embryos at different time points of development with TRIzol reagent (Thermo Fisher Scientific), and cDNA samples were synthetized with SuperScript II Reverse Transcriptase (Invitrogen). Quantitative PCR was performed in a LightCycler PCR instrument (Roche) using SYBR Green Master Mix (Roche). Reaction conditions were carried out for 50 cycles (10 min Initial denaturation 95 °C, 10 s at 95 °C, 15 s at 60 °C and 20 s at 72 °C), each qRT-PCR reaction was performed in triplicate and relative mRNA expression was normalized to GAPDH for Tubg1 and Tubg2 comparisons and to Actin for mouse models.

The following gene-specific primers were used for mouse γ-tubulin 1: Forward: 5'CCCAGGGAGAAAAAATCCACGAGGA3'; Reverse: 5'GAGCCCAAGCCAGAGCCTGTCC 3'; for mouse γ-tubulin 2: Forward: 5'GACCGAGAAGCAGATGGAG3'; Reverse: 5'CGTTCAGTCGCTCTAAGAGG3'.

For mouse model validation, the following primers were used between exons 1 and 4 for KI mice: Forward 5'GAGGAGCGATGCCGAGAGAA3', Reverse 5'CATCTGCCTCCCGATCTATG3' and between exons 10 and 11 for KO mice: Forward: 5'TCTCCTCGCTCTTCGAACGGA3' and Reverse: 5'TCAATCAGCTGCTGCACAAT3'. The presence of the mutation in KI mice was verified by PCR on cDNA samples using Taq DNA Polymerase (Roche) using the same primers as those used for qPCR. PCR products were sent to GATC biotech for Sanger sequencing.

**Immunohistochemistry and immunofluorescence**. Free floating sections were blocked in 2% donkey serum in PBS-0.3% Triton-X and then incubated overnight at 4 °C with the primary antibodies. Alexa-coupled secondary antibodies (Thermo Fischer) were used at 1/500. Nuclei were stained with a Hoechst solution (H3570, Life Technologies) for 20 min, used at 1/2000.

Neuro2a cells grown on glass coverslips were fixed in 4% PFA, blocked in 5% donkey serum in PBS-0.3% Triton-X and then incubated for 1 h with the primary antibodies. Alexa-coupled secondary antibodies (Thermo Fischer) were used at 1/500.

Human fibroblasts grown on glass coverslips were fixed in 3% PFA-PHEM buffer (60 mM Pipes, 25 mM Hepes, 5 mM EGTA, 1 mM MgCl, pH 7), post-permeabilized in 0.5% Triton X in PHEM for 4 min at 37 °C and post-fixed in ice cold methanol for 2 min on ice. Fibroblasts were then blocked in 5% donkey serum in PHEM-0.3% Triton-X and then incubated for 1 h with the primary antibodies. Alexa-coupled secondary antibodies (Thermo Fischer) were used at 1/500.

Images were acquired using a confocal microscope TCS SP8 X (Leica microsystems).

**In utero electroporation**. In utero electroporation was performed using timed pregnant Swiss mice (Janvier)[62]. Animal experimentations were performed at the IGBMC animal facilities. The study was conducted according to national and international guidelines (authorization numbers 2017062811273521 and 2017022316297963, French MESR) and the procedures followed were in accordance with the ethical standards of the responsible committee on mouse experimentation (Comité d'éthique pour l'expérimentation animale (Strasbourg, France)). At E14.5 the pregnant mice were anaesthetized with isoflurane ($2 \, l \, min^{-1}$ of oxygen, 3% isoflurane during induction and 2.5% during surgery). The uterine horns were exposed, and a lateral ventricle of each embryo was injected using pulled glass capillaries (Harvard Apparatus) with Fast Green ($2 \, \mu g \, ml^{-1}$; Sigma) combined with the DNA constructs prepared with EndoFree plasmid purification kit (Macherey Nagel) and a reporter vector at $1 \, \mu g \, \mu^{-1} l$ each. Plasmids were further electroporated into the neuronal progenitors adjacent to the ventricle by delivering five electric pulses at 50 V for 50 ms at 950 ms intervals using a CUY21EDIT electroporator (Sonidel). After electroporation, embryos were placed back in the abdominal cavity and development was allowed to continue until E16, E18, P3 ot P8. Embryo or pup brains were dissected and fixed in 4% paraformaldehyde in PBS (Prosphate buffered saline) overnight and then sectioned at 80 μm slices using a VT1000S vibratome (Leica biosystems).

**Time-lapse videomicroscopy**. For time-lapse videomicroscopy, E16.5, embryo brains electroporated 2 days earlier, were dissected, embedded in 3% low-melt agarose (BioRad) diluted in HBSS (Hank's Balanced Salt Solution, ThermoFisher Scientific) and sliced (300 μm) with a vibratome (Leica VT1000S, Leica Microsystems)[63]. Brain slices were cultured 16–24 h in semi-dry conditions (Millicell inserts, Merck Millipore), in a humidified incubator at 37 C in a 5% CO2 atmosphere in wells containing Neurobasal medium supplemented with 2% B27, 1% N2, and 1% penicillin/streptomycin (Gibco, Life Technologies). Slice cultures were placed in a humidified and thermoregulated chamber maintained at 37 °C on the stage of an inverted confocal microscope. Time-lapse imaging was performed with a Leica SP8 X scanning confocal microscope equipped with a 25X objective. Forty to fifty successive 'z' optical planes were acquired every 0.5 μm and every 30 min during 10 h. Sequences were analyzed using Image J and the 'Manual Tracking' Plugin.

**EdU labelling**. E14.5 electroporated pregnant mice were intraperitoneally injected 24 h later with EdU (5-ethynyl-2′-deoxyuridine solution, Invitrogen) diluted in sterile normal saline at the dose of 40 mg EdU kg$^{-1}$ of body weight. Brains from embryos were dissected and fixed at E16.5. EdU staining was performed using Click-iT EdU Alexa Fluor 647 Imaging Kit (Invitrogen) according to the manufacturer's instructions.

**Mouse model and genotyping**. Heterozygous Tubg1 knock-out and knock-in mice expressing the Tyr92Cys ($Tubg1^{Y92C/+}$) variant were generated in the Institut Clinique de la Souris (Celphedia, Phenomin, ICS, Illkirch). The targeting vector (shown in Supplementary Fig. 6) was constructed as follows. An 843 bps fragment encompassing part of Tubg1 exons 2 and 3 was amplified by PCR (from RP23-440M18 BAC containing Tubg1) in two steps to allow the introduction of the A > G point mutation in exon 3 and subcloned in an ICS proprietary vector. This mutation changes the TAC codon in a TCC codon leading in a tyrosine to cysteine mutation at position 92 (Y92C). The ICS vector has a flipped Neomycin resistance cassette as well a 2 LoxP sites, one is located 5′ of the 843 bps fragment and NeoR cassette and one 3′ of the flipped NeoR cassette. Two fragments corresponding to the 5′ (3.5 kb) and 3′ (3.5 kb) homology arms were amplified by PCR and subsequently subcloned in step1 plasmid to generate the final targeting construct. The linearized construct was electroporated in C57BL/6N mouse embryonic stem (ES) cells (ICS proprietary line). After G418 selection, targeted clones were identified by long-range PCR and further confirmed by Southern blot with an internal (Neo) probe and a 3′ external probe. Two positive ES clones were validated by karyotype spreading and microinjected into BALB/C blastocysts. Resulting male chimeras

were bred with Flp deleter females showing maternal contribution[64]. Germline transmission with the direct excision of the selection cassette was achieved in the first litter. The initial allele comprising the floxed and mutated fragment represents the KI allele. The KO allele can be obtained subsequently by breeding the initial allele with a Cre deleter line. Genotyping was performed by standard short-range PCR, using the following primers: Lf, CCTCGCAATAAGGTCATAGCTTCGG; Mf, GTGATCCATTCCATCCTAAACTCCTC; Ef, GGAACGCTAAGGACGCA GTCTAG; Mqf, GAAGAACGAGATCAGCAGCCTCTGTTCC; Lxr, CATACAT TATACGAAGTTATCTGCAG; Lr, GGTAGGTTCCAAATAGCCAGTGGG; Mr, CTGTGCCCACCCACAGAGTTGAG; Mqr, TGCTAAAGCGCATGCTCCAGA CTGC and Er, GCCTTCCTTCTCCATTGCGCATTG. The position of the different primers used are represented in Supplementary Fig. 6a. To determine the genotype, the presence and size of PCR fragments were analyzed to detect: the presence of exon 3 (Mf/Mr); the presence of distal LoxP (Lf/Lr), excision of the selection marker (Ef/Er), 5′ part of the selection marker (Ef/Mqr); 3′ part of the selection marker (Mqf/Er); LoxP specific PCR (Lf/Lxr) and the excision of the floxed exon (Lf/Er) The presence of the point mutation was confirmed by Sanger sequencing on the first F1 animals.

Mice were handled according to national and international guidelines (authorization number 2017022316297963, French MESR) and the procedures followed were in accordance with the ethical standards of the responsible committee on mouse experimentation (Comité d'éthique pour l'expérimentation animale (Strasbourg, France)).

**Neuroanatomical studies**. Adult neuroanatomical studies were carried out using eight male mice ($n = 4$ per group) for Tubg1 knock-in mice and 12 mice ($n = 3$ per group, male and female) for Tubg1 knock-out mice. All mice were precisely 16-weeks-old. Mice were euthanized in a CO2 chamber and brains were dissected and fixed in 4% buffered formalin for 48 h, then transferred to 70% ethanol. Samples were embedded in paraffin using an automated embedding machine (Sakura Tissue-Tek VIP) and cut at a thickness of 5 μm with a sliding microtome (HM 450; Microm) in order to obtain sagittal brain region at Lateral +0.60 mm. The sections were then stained with 0.1% Luxol Fast Blue (Solvent Blue 38; Sigma-Aldrich) and 0.1% Cresyl violet acetate (Sigma-Aldrich) and scanned using NanoZoomer 2.0HT, C9600 series at 20x resolution. 166 brain parameters, made of area and length measurements as well as cell level features, were taken blind to the genotype across one precise sagittal section at Lateral +0.60 mm for Tubg1 knock-in mice. For Tubg1 knock-out mice, we measured 67 parameters across two coronal planes at Bregma +0.98 and Bregma −1.34 mm. Cortical layers were visually assessed based on neuronal differences in cytoarchitecture. Data were analyzed using a two-tailed Student t-test of equal variance to determine whether a brain region is associated with neuroanatomical defect or not. Similarly, neuroanatomical studies were carried out at postnatal day 0 ($n = 5$ WT and $n = 4$ Tubg1$^{Y92C/+}$ mice). Brain sections were cut exactly in the plane of the closest image to the section defined as critical section at Lateral +0.60 mm of the right hemisphere. Forty brain parameters, made of area and length measurements, were taken blind to the genotype.

**Behavioral experiments**. All mouse experimental procedures were performed in agreement with the EC directive 2010/63/UE86/609/CEE for the care and use of laboratory animals and every effort was made to minimize the number of animals used. The local animal care, use and ethic committee of the Institute of Genetics and of Molecular and Cellular Biology (IGBMC) approved the protocol under accreditation number 2017022316297963. One cohort of wild-type and mutant male mice were used for behavioural experiments (nine wt and nine Tubg1$^{+/-}$ in total, from 2.5 months ±1 week of age for the phenotyping). For all behavioural experiments, mice were kept in SPF conditions with free access to food and water. The light cycle was controlled as 12 h light and 12 h dark (lights on at 7:00 AM) and the tests were conducted between 9:00 AM and 4:00 PM. To produce experimental groups, only animals coming from litters containing a minimum of two male pups were selected. After weaning, male mice were gathered by litters in the same cage. Animals were transferred to the experimental room 30 min before each experimental test. Behavioural experimenters were blinded as to the genetic status of the animals. Tests were administrated in the following order, circadian activity, Y maze, OF, NOR, repetitive behaviours and FC with a maximum of one week in between two tests.

For the OF, the apparatus consisted of a white circular arena (52 cm diameter, 32 cm height) placed in a dimly lit testing room (30–40 lux). The mice were placed for 15 min in this arena for two successive sessions with 24 h of interval. During the sessions, mice were monitored using a video tracking system (Ethovision, Wageningen, The Netherlands), total distance travelled and time spent in the different zone of the arena (center, intermediate and peripheral) was recorded.

For circadian activity, spontaneous locomotor activity and rears were measured using individual boxes equipped with infra-red captors. Mice were tested for 45 h in order to measure habituation to the apparatus (two dark phase) as well as nocturnal and diurnal activities, represented by locomotor and rear activity. Results are expressed per 1 h periods.

For studying repetitive behaviour, mice were put individually into dimly lit (60 Lux) clean home-cages without pellets or water bottle. The occurrence of repetitive behaviours (rearing, jumping, digging, grooming) was observed for 10 min and scored using an ethological keyboard (Viewpoint, Labwatcher, France). This test was done only a cohort of nine WT and nine KI animals.

The NOR task was done 24 h after OF the session performed in the same apparatus. On Day 1, mice were placed in the test apparatus containing two identical objects and were free to explore them for 14 min (defined as a training session). The time spent exploring each object was manually scored and nose to the object at a distance <1.5 cm. After this acquisition phase, mice were returned to their home cage. After a waiting period of 24 h, mice were placed once again in the test apparatus with one FO (from the acquisition phase) and one NO. Mice were again free to explore the two objects for a 14 min period (defined as a test session) and time spent exploring each object was manually scored. Between trials and subjects, the different objects were cleaned with 70° ethanol in order to reduce olfactory cues. To avoid a preference for one of the two objects, the new one was different for the different animal groups and counterbalanced between genotype. It was also the case for the emplacement of the NO compared to the familiar one (left or right). For the retention phase, a RI was calculated as the percentage of time sniffing time NO minus the time sniffing the FO divided by the total time of sniffing, to assess memory performance. For this test, four WT mice were excluded because they did not come into contact with objects (sniffing time lower than 3 s).

FC (Contextual and Cued FC) experiments were conducted in four operant chambers (28 × 21 × 22 cm) with a metal bar floor linked to a shocker (Coulbourn Instruments, Allentown US). Chambers were dimly lit with a permanent house-light and equipped with a speaker for tone delivery and an infra-red activity monitor. The activity/inactivity behaviour was monitored during the different session and data were expressed in duration of inactivity per 2 s. The experimental procedure consists of three sessions over 2 days. On day 1, for the conditioning session, the mouse was allowed to acclimate for 4 min, then a light/tone (10 kHz, 80-dB) conditioned stimulus (CS) was presented for 20 s and terminated by a mild foot shock unconditional stimulus (US) (1 s, 0.4 mA). After the foot shock, animals were left in the chamber for another 2 min. Total freezing time during the first 2 min and 4 min and 2 min immediately after foot shock was defined, respectively, as PRE1, PRE2 and POST. On Day 2, the context testing was performed by putting back the mice into the same chamber and allowing them to explore for 6 min without presentation of the light/auditory CS. The movement of the animal were monitored to detect freezing behaviour consequent to recognition of the chamber as the spatial context (contextual learning). The total freezing time was calculated by 2 min block as CONT2, CONT4 and CONT6. Finally, the cue testing was performed 5 h after the context testing. Animals were tested in modified conditioning chambers with walls and floor of a different colour and texture. The mouse was allowed to habituate for 2 min then presented to light/auditory cues for 2 minutes to evaluated conditioning fear. The total freezing time was calculated by 2 min block as PRECUE1, CUE1, PRECUE2, and CUE2.

The Y maze was made of three enclosed plastic arms, each 40 × 9 × 16 cm, set at an angle of 120° to each other, in the shape of a Y. The wall of the arm has different pattern to encourage spontaneous alternation behaviour (SAB). Animals were placed at the end of one arm (this initial arm was alternated within the group of mice to prevent bias of arm placement), facing away from the centre, and allowed to freely explore the apparatus for 6 min under moderate lighting conditions (70 lux in the centre-most region). The time sequences of entries in the three arms were recorded (arm entry was counted when mouse had all four paws inside the arm). Alternation was determined from successive entries of the three arms on overlapping triplet sets in which three different arms are entered. The number of alternations was then divided by the number of alternation opportunities namely, total arm entries minus one. In addition, total entries were scored as an index of locomotor activity.

Data were assessed for normality by Shapiro–Wilk test and for equality of variance by Brown–Forsythe test. If normality and equality of variance is assumed, we performed a one-way ANOVA analysis. In case of normality or equal of variance failure, we performed a one-way ANOVA on the rank (Kruskal–Wallis). By the same way, data from OF, NOR, and FC were assessed by repeated measure two-way ANOVA with genotype as between subject and session as within subject. All pairwise comparison was done with Tukey test in post hoc analysis. Also NOR RI was analysed with one sample t-test for determined a significant difference versus 0.

**EEG recordings**. Implantation of electrodes was done under general anesthesia (Propofol, Fentanyl, Domitor mixture 10 µL g⁻¹). Heterozygous $Tubg1^{Y92C/+}$ ($n = 3$) and C57Bl/6 wild-type littermate mice ($n = 3$) aged postnatal 12 weeks were implanted with stainless steel wire electrodes (Phymep, France). For each mouse, five single-contact electrodes were placed over the left and right fronto-parietal cortex. The electrodes were secured into the skull and soldered to a micro-connector that was fixed to the skull by acrylic cement. An electrode over the surface of the cerebellum served as ground for all derivations. All mice were allowed to recover for a period of 1 week before EEG recordings. Freely moving mice were then recorded in their housing transparent cage and were connected to a recording system. EEG signals were amplified with a band-pass filter setting of 0.1–70 Hz with a 64-channel system (Coherence, Natus) and sampled at 256 Hz. Recordings were performed during 1 h for evidence of spontaneous convulsive seizure. At the end of the recording period, animals were injected intraperitoneally with a convulsive dose of 30 mg kg⁻¹ of pentylenetetrazole (PTZ; Sigma-Aldrich, Co), a GABA$_A$ receptor antagonist, to evaluate seizure threshold. EEG was recorded during 30 min after PTZ. Video-EEGs were reviewed offline for electro-graphic seizures.

**Quantifications and statistical analysis**. All experiments were done in at least three independent replicates and analysis were performed blinded to genotype or condition. All statistics were calculated with GraphPad Prism 6. Data are presented as mean ± sem. The experimental n, the test used and the statistical significance can be found in the figure legends. For in utero electroporation experiments cell countings were done in three different brain slices of at least three different embryos, pups or mice for each condition. Time-lapse recordings were performed in a minimum of three electroporated embryos per group. For all histological quantifications performed in $Tubg1^{Y92C/+}$ and $Tubg1^{+/-}$ mice three brain sections per brain of at least three different animals per condition were analyzed. Morphometric analysis of the knock-in and knock-out mice, a minimum of three animals were studied and analyzed in a single specific brain section. Experiments performed in cells (N2A, HeLA, Fibroblasts) were done independently at least three times. For EB3 comets, in each experiment 10 cells were analyzed and in each cell 15–30 comets were quantified. For re-polymerization, in each experiment around 100 cells were analyzed per condition. For behavioral analysis, 9 male mice of each genotype were used. Three mice of each genotype were used for EEG recordings and challenged with PTZ. A more detailed description of statistics is provided in Supplementary Dataset 3.

**Reporting summary**. Further information on research design is available in the Nature Research Reporting Summary linked to this article.

## Data availiability

The source data underlying Figs. 1c; 2b, c; 3b–g, i–k; 4b, c, e, g–i; 5b, d, g–m; 6a, c; 7a–e; and Supplementary Figs. 1c, d; 2a–c, e, g, i; 3a, c–f; 4d; 5c, d; 6d–f; 7; 8; 9a; 10a–j, l–o; 11b–e are provided as a Source Data file. All other relevant data included in the article are available from the authors upon request.

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

## Acknowledgements

This work was supported by grants from the EU Seventh Framework Programme FP7 under the projects GENCODYS (COLLABORATION PROJECT-2009-2.1.1-1/241995 to H.S., Y.H. and J.C.) and DESIRE (grant agreement no. 602531 to J.C), the French state funds through the Agence Nationale de la Recherche˙ under the projects ANR-15-CE16-0018-01, and the programme Investissements d'Avenir labelled [ANR-10-IDEX-0002-02, ANR-10-LABX-0030-INRT, to B.Y., J.G, J.C., Y.H., and ANR-10-INBS-07 PHENOMIN to Y.H.]. The funders had no role in the study design, data collection and analysis, decision to publish or manuscript preparation. This work was also supported in part by Grants 16–23702s (Vadym Sulimenko) and 18–27197s (Pavel Dráber) from the Grant Agency of the Czech Republic and by Institutional Research Support (RVO 68378050). We thank the Imaging Center of IGBMC (ici. igbmc.fr), in particular Basile Gurchenkov, Elvire Guiot and Erwan Grandgirard for their assistance in the microscope imaging experiments and Nadia Messaddeq from the electron microscopy platform for the data generated. We are grateful to the staff of the mouse facilities of the Institut Clinique de la souris (ICS) and Institut de Génétique et de Biologie Moléculaire et Cellulaire (IGBMC) for their involvement in the project. We also thank the lab of David Agard (San Francisco, California), and in particular Axel Brilot for their assistance and suggestions. We thank Maria Osipenko for the analysis of cortical thickness in adult knock-in mice. We are grateful to Julien Couchet for providing us the pCAG-mScarlet and pSilencer-CAG-Venus-2 plasmids. We would like to acknowledge Mohammed Selloum and Hamid Meziane for their implication in the characterization of Tubg1$^{+/-}$ model.

## Author contributions

E.L.I. conceived and designed the experiments, performed experiments and analyzed the data of IUE, Time-lapse, cellular studies in fibroblasts and N2a cells, immunohistochemistry studies in mice and wrote the manuscript. J.G.G. performed and analyzed EB3 studies in fibroblasts and immunohistochemistry studies in mice and reviewed the manuscript. Vad.S. conceived, designed, performed and analyzed the data of immunoprecipitation studies and reviewed the manuscript. A.D. performed and analyzed behavioral studies. G.R. performed and analyzed EEG studies. K.R. participated in IUE and Time-lapse studies. S.C.C. performed neuroanatomical studies. L.A. provided assistance with Time-lapse studies. L.B. provided assistance and contributed with suggestions. N.D, and P.T. provided technical assistance and prepared reagents. P.N. provided patient fibroblasts. W.M. and A.V. and Val.S. provided technical assistance and advice for in vivo studies in mice. J.D.G. provided assistance, contributed with suggestions and reviewed the manuscript. M-C.B. and G.P. participated in the development of the mouse model. B.Y. designed experiments related to neuroanatomical studies. Y.H. designed experiments related to behavioral studies and reviewed the manuscript. P.D. designed experiments related to immunoprecipitation studies, provided assistance, contributed with suggestions and reviewed the manuscript. J.C. conceived, coordinated and supervised the study. M-V.H. conceived, designed, performed and analyzed the data of proliferation and EB3 in HeLa cells studies, coordinated and supervised the study.

## Additional information

**Competing interests:** The authors declare no competing interests.

