## [Peer Review File · Nature Communications]

Reviewers' Comments:

Reviewer #1:

Remarks to the Author:

Here, the authors characterize four patient specific missense mutations in the gamma tubulin gene TUBG1 that are linked to malformations of cortical development, by using overexpression of the variants in embryonic mouse neocortex via in utero electroporation, patient-derived fibroblasts and a Tubg1Y92C/+ knock-in mouse model. Phenotypic analysis was conducted by overexpressing the pathogenic variants in embryonic mouse neocortex under the control of a strong mammalian expression promoter CAG, and a neuron-specific promoter for Doublecortin (DCX). While the overexpression led to altered migration of postmitotic neurons, as assessed by fixed and live imaging, progenitor proliferation was not affected. Experiments using patient derived fibroblasts and HeLa cells revealed that TUBG1 variants decreased microtubule dynamics and have reduced ability to form gamma tubulin complexes. Finally knock-in mutant mouse model was generated which showed neuroanatomical and behavioral defects consistent with the human phenotype, suggesting that it is a useful model of further investigation of human pathology.

This is the first study involving in vivo characterization of patient specific TUBG1 variants in embryonic mouse brain and therefore potentially very interesting. Since the variants affect microtubule dynamics, it follows that the cellular processes that depend on microtubules such as progenitor proliferation and migration would be affected. Authors should explain why the variants affect post mitotic neuronal migration but not proliferation of radial glial progenitor cells. This is important because the patients as well as the mouse model exhibit microcephaly – a phenotype associated with reduced proliferation.

Since gamma tubulin complexes are known to regulate microtubule dynamics, the claim that altered neuronal migration results from altered microtubule dynamics, independently of centrosome integrity and function may not be entirely correct.

The authors should explain why Tubg1Y92C/+ was specifically selected for generating the mouse model over other mutations.

Suggestions

1) Results obtained with experiments where a given protein is overexpressed should be interpreted with caution. This is because even a wild type protein expressed above its normal physiological expression level can potentially produce a phenotype due to toxic gain of function (e.g. Figure 4 E). Therefore Tubg1 RNAi rescue by expression of RNAi resistant wild type or mutant Tubg1 should be performed at least in a small number of experiments. Also, the results of variant overexpression should be compared to those of wild type Tubg1 overexpression (as opposed to empty vector expression) with statistical tests, where possible.

2) Why are different promoters used to drive the expression of a specific variant (DCX) and the reporter (NeuroD) in post mitotic neurons? This is also the case with experiments in radial glial progenitor cells (CAG promoter for driving the expression of a variant and BLBP for the reporter).

3) Time-lapse images or videos for should be provided for live imaging of neuronal migration since these were used for analyzing the velocity, number of pauses, pause duration in migrating neurons. Similarly, images of EB3 comets should be provided along with representative kymographs.

4) The authors claim to address the functional redundancy of Tubg1/2 by showing a partial rescue of

phenotype produced by mutant form of Tubg1 by overexpression of TUBG2. However this is experiment does not address the redundancy between the two genes. It would be addressed by the rescue of Tubg1 knockdown phenotype by Tubg2 overexpression or vice versa.

5) Expression of the mutant forms of Tubg1 leads to the arrest of neurons in the intermediate zone. The arrested neurons show multipolar morphology. However the defect in the multipolar to bipolar transition should be quantified.

Reviewer #2:

Remarks to the Author:

In this manuscript the authors explore the cellular and molecular defects that are associated with TUBG1 mutations and cortical malformations. They have employed a wide range of sophisticated methods including: in utero electroporation; live cell time lapse imaging of migrating neurons; cellular analysis of microtubule dynamics; the generation of in vivo mouse models; and EEG analysis. They have made a number of important findings - which focus on defects in post-mitotic neurons rather than the progenitors themselves. This (I think) is unexpected and interesting. In general the paper is well written, easy to follow, and makes a valuable contribution to the field.

I think this paper merits publication in Nature communications, but first requires some work. While the manuscript has numerous strengths in its current form there are number of aspects that I found deeply frustrating. First, the authors have consistently failed to specify the n numbers in the figure legends nor have they used dot plots. Accordingly, its very difficult for the reader to assess whether their data really does justify their conclusions. Secondly, their mouse model has been analyzed superficially, despite being the punch line of the abstract. Finally, there are a number of experiments which require more information with respect to the controls used and how they were quantitated.

Accordingly, I would suggest the following improvements.

Major Issues:

1. The authors should specify in the main text what control was used for their fibroblast experiments. The control used is critical to the interpretation of their patient data. Were the controls parental fibroblasts? This would be ideal. A single control from an unrelated individual is not sufficient to draw any conclusions (3 unrelated control individuals is a minimum). Moreover, the authors should quantitate the "shorter and less organized" microtubules following depolymerisation, rather than relying on qualitative assertions.

2. Further validation of their mouse model is required. Specifically, the authors should performed western blot analysis at an appropriate developmental time point to assess the levels of TUBG1 protein in their Y92C/+ animals, as well as the knockout animals. Is there a truncated protein in their KO animals? They should likewise assess the levels of mRNA in these animals.... is there nonsense mediated decay in their KO animals? Whereabouts is the termination mutation in the KO animals?

3. The analysis of their mouse mutants is superficial and currently under utilizes this valuable resource. The authors have not analyzed the cortex in sufficient depth for their Y92C/+ animals or their KO animals. An analysis of lamination in adult animals should be performed (e.g. Foxp2, ER81, Cux1), and cortical thickness assessed. Is the number of interneurons similar?

4. The authors should assess neurogenesis in their mouse mutants, reflecting the experiments they have performed in utero. In vivo models are always better, and it would be valuable to confirm that neurogenic output is similar in their mice. An assessment of PH3 at E14.5 for instance would be valuable, as well as the number of SOX2 positive progenitors. Stating in the discussion "data now shown" is not good enough. The words "data not shown" should not appear in any scientific manuscript.

5. The methods should include a section that describe how sections were quantitated. Moreover, all quantification should be performed blinded to genotype or condition. If this has not been done, the data should be re-analyzed.

6. I would restructure the presentation of their mouse data - beginning with the overall volumetric assessment of the brain regions. From there you could then focus on the cortex, hippocampus and cerebellum in a sequential manner. Building up Figure 5, and leaving Figure 6 just for behavior and EEG.

7. Figure 6 presents the results of their behavioral experiments. I have some concerns about this. Were sex matched littermate controls used for their experiments? It is strange that 19wt animals and 17 TUBG1 were used. Were all experiments performed blind to genotype? Moreover, if indeed the ages ranged from 2.5 to 6.5 months of age this is a major concern. Only data from animals of the same age should be shown (+/- a week). If there is a degenerative aspect to this phenotype - using animals of different ages is problematic.

Review of Ivanova:

Minor issues:

1. The authors do not indicate the n number for the in utero electroporation experiments in either the figure legends, text or methods. You cannot expect your reader to download a supplementary table for this information. Please rectify this. How many sections were analyzed for each animal? Again this is not specified.

2. The Figure legends for supplementary figure 1, do not actually match supplementary figure 1. Should the first panel be deleted?

3.. The n number for the PH3 experiments shown in Figure S2 is not specified. Again, this requires rectification. Moreover, all data should be presented as dot bar graphs so the reader can better appreciate the variation in the data set. An image showing the PH3 staining should also be included.

4. The authors state on page 9 lines 272 that "Our results indicate that disrupted locomotion in neurons expressing recombinant TUBG1 is not due to defects in centrosome positioning". This statement requires some clarification, as it might be interpreted as WT recombinant TUBG1..... and given the centrosomal phenotype of those mice expressing TUBG1-Tyr92Cys this assertion is confusing.

5. The n numbers for comets and cells analyzed for the microtubule dynamics experiments should again be specified in the figure legends for Figure 4, and again dot and bar graphs employed. How many different times was this experiment replicated? Were they independent transfections?

6. On panel H of Figure 4, the TUBG1-Leu387Pro pull-down results in a noticeably lower amount of Gamma tubulin.... while the TAG-RFP seems similar to the other constructs. How do the authors explain this?
7. With respect to Figure 4, it would be useful to show the full western blot gels with a ladder in the supplementary data.
8. Please expand supplementary Figure 5, showing the southern blot results demonstrating correct homologous recombination in ES cells. How were animals subsequently genotyped?
9. What Cre line was employed to generate KO animals?
10. The TBR1 staining shown in Figure 5A is not informative. Layer VI? Everything seems to be stained.
11. Again no n numbers shown in Figure legend 6.

Reviewer #3:

Remarks to the Author:

It has been known that TUBG1 variants in patients link to cortical dysgeneses associated with intellectual disability and epilepsy. However, the underlying mechanisms remain elusive. In this manuscript, Ivanova and colleagues investigate four human MCDs-related TUBG1 missense variants on cortical development by using in-utero electroporation and a knock-in *Tubg1*(Y92C/+) mouse model. They found that TUBG1 variants affect neuronal migration but do not effect on progenitor proliferation. Using time-lapse videomicroscopy, authors further demonstrated TUBG1 variants affect microtubule dynamics rather than centrosomal nucleation ability. In addition, they reported that *Tubg1*(Y92C/+) mouse showed neuroanatomical and behavioral defects.

Although the manuscript clearly represents a significant amount of work and contains interesting data, I have some suggestions/concerns.

1. To highlight the significance of this manuscript, it's necessary to further emphasize these four TUBG1 variants are human MCDs-related variants, and cite relative literatures.
2. Although the typical image data (Fig. S1D) show that GFP+ cells electroporated TUBG1 WT and four TUBG1 variant vectors overexpress TUBG1, authors should further quantify the percentage of overexpressed GFP+ cells in GFP+ cells in each group.
3. Please explain why quite number of *Cux1*+/*RFP*- cells in four TUBG1 variants locate in the lower layer of neocortex (Fig. 1D and 1E). Moreover, to exclude the possibility that mislocalized cells co-express lower layer markers, *Ctip2* and *Tbr1* markers should be tested in these cells. Similarly, in DCX promoter experiments (Fig. 2), co-expression of *Cux1*, *Ctip2* and *Tbr1* markers with electroporated neurons should be examined.
4. Through the quantifications (Fig. 1C and Fig. 2B), we can observe the phenotype of CAG promoter is much severe than that of DCX promoter. Please discuss possible reasons.
5. To test the neuronal mispositioning is due to an arrest rather than a delay in neuronal migration, authors quantified the distribution of electroporated neurons in postnatal stages, CAG promoter at P8 and DCX promoter at P3. However, as compare with neuronal distribution at E18 in DCX promoter experiments (Fig. S3B), electroporated neurons can still migrate at P3 (Fig. 2B). Therefore, I suggest to quantify the distribution of electroporated neurons for both CAG and DCX promoter experiments at relative mature postnatal stage (e.g., P20).
6. In rescue experiments, co-electroporation of TUBG1 or TUBG2 can partially rescue Tyr92Cys TUBG1 variant-induced positioning phenotype. It will be very helpful to test whether co-electroporation of

TUBG1 or TUBG2 can rescue other TUBG1 variant-induced phenotypes. In addition, authors mentioned that "Co-electroporation of either isoforms leads to a rescue in the neuronal positioning phenotype in a dose-dependent fashion". Although it is correct for TUBG2, the rescue in the neuronal positioning phenotype for TUBG1 seems not to be in a dose-dependent fashion, since there is no significant change for neuronal positioning between two TUBG1 electroporated concentrations (Fig. S1H).

7. Though current data, it's really hard to conclude "variants affect neuronal migration dynamics but not centrosomal positioning". In other words, there is no direct evidence to support Tyr92Cys variant do not affect centrosomal positioning and further influence the neuronal migration. In order to demonstrate that Tyr92Cys variant still influences the neuronal migration even under centrosome migrating position, authors should further compared the percentages of migrating neurons and non-migrating neurons whose centrosome was positioned ahead of the nucleus between control-empty vector and DCX-TUBG1-Thr331Pro groups.

8. For Tubg1(Y92C/+) mouse, the phenotypes are much weaker than CAG and DCX promoter electroporation experiments. I suggest to systematically quantify the distribution of cortical layers in Tubg1(Y92C/+) mouse at relative mature postnatal stage (e.g., at P20). Moreover, it's worthy to further examine the neuronal migration and centrosome positioning in Tubg1(Y92C/+) mouse at embryonic stage through EGFP vector co-electroporation.

9. Authors should discuss why Leu387Pro variant does not affect the microtubule dynamics, but severely influence the neuronal positioning.

10. It is nice to find mutant Tubg1(Y92C/+) mice show an increased epileptic cortical activity. However, the more detail information should be included. For example, how many Tubg1(Y92C/+) mice showed epileptic cortical activity.

Minor Comments:

1. There are too many "data not shown" in this manuscript. At least, authors should provide the original data that Tubg1+/- mice showed no major abnormalities in their neuroanatomy and no behavioral alterations.

2. Authors should provide the image data that stained PH3, Pax6, Tbr2, and EGFP/RFP.

3. Fig. S1A is redundant, since it has been already shown in Fig. 1A. Moreover, the number of the supplementary Figure 1 marked in alphabetical order in main text and figure legend is incorrect. Please revise.

4. Authors electroporated pCAGGS-IRES-Tomato reporter, but showed GFP in Fig. S1D (should be Fig. S1C). Please check this.

5. In line 394, "μ-aminobutyric acid" should be "γ-aminobutyric acid".

Dear reviewers,

We would like to thank you for the encouraging and constructive comments regarding our manuscript. We carefully considered all your comments and suggestions during the revision process and would therefore like to acknowledge you for helping us improve our manuscript.

We apologize for the time it took us to finalize revisions, but we would like to highlight that this is mainly due to the additional experiments requiring time-pregnant mice and in utero electroporations. The new version of the manuscript includes now a number of new experimental data (listed below) and was revised accordingly:

- Electron microscopy showing no structural defects in centrosomes
- EB3 comets dynamics using 3 different control fibroblasts
- MT depolymerization assay using 3 different control fibroblasts
- Tubg1 sh-RNA rescue with wild type and mutant TUBG1 by in utero electroporation
- Tubg1 sh-RNA rescue by TUBG2 overexpression using in utero electroporation
- Wild-type TUBG1 and TUBG2 rescue for all TUBG1 pathogenic variants by in utero electroporation
- Analysis of the distribution of CAG and DCX TUBG1 electroporated neurons at P8 and P21
- Validation of the TUBG1 heterozygous knock-in and knock-out mouse models
- The morphometric analysis of newborn TUBG1^{Y92C/+} knock-in mice brains
- Lower and upper-layers markers expression in neurons electroporated with either CAG and DCX driven TUBG1 variants
- Proliferation parameters during cortical development in the TUBG1^{Y92C/+} knock-in mouse model (E12, E14 and E16)
- The neurogenic output in the TUBG1^{Y92C/+} knock-in mouse model
- Cortical layering during corticogenesis in TUBG1^{Y92C/+} knock-in mouse model
- Lamination in adult TUBG1^{Y92C/+} brains
- Identity of EdU mislocalized neurons in the TUBG1^{Y92C/+} knock-in mouse model
- Video microscopy recordings and time-lapse analysis of morphology transition and centrosome positioning during migration process in the TUBG1^{Y92C/+} knock-in mouse model.

We believe that with these new results and additional analysis performed we have now fully addressed each point raised by the reviewers. We hope you will find this revised version of the manuscript is now acceptable for publication.

Sincerely,

Dr. Maria-Victoria HINCKELMANN
Pr. Jamel CHELLY

Reviewers' comments:

Reviewer #1 (Remarks to the Author):

Here, the authors characterize four patient specific missense mutations in the gamma tubulin gene TUBG1 that are linked to malformations of cortical development, by using overexpression of the variants in embryonic mouse neocortex via in utero electroporation, patient-derived fibroblasts and a Tubg1Y92C/+ knock-in mouse model. Phenotypic analysis was conducted by overexpressing the pathogenic variants in embryonic mouse neocortex under the control of a strong mammalian expression promoter CAG, and a neuron-specific promoter for Doublecortin (DCX). While the overexpression led to altered migration of postmitotic neurons, as assessed by fixed and live imaging, progenitor proliferation was not affected. Experiments using patient derived fibroblasts and HeLa cells revealed that TUBG1 variants decreased microtubule dynamics and have reduced ability to form gamma tubulin complexes. Finally knock-in mutant mouse model was generated which showed neuroanatomical and behavioral defects consistent with the human phenotype, suggesting that it is a useful model of further investigation of human pathology.

This is the first study involving in vivo characterization of patient specific TUBG1 variants in embryonic mouse brain and therefore potentially very interesting. Since the variants affect microtubule dynamics, it follows that the cellular processes that depend on microtubules such as progenitor proliferation and migration would be affected. Authors should explain why the variants affect post mitotic neuronal migration but not proliferation of radial glial progenitor cells. This is important because the patients as well as the mouse model exhibit microcephaly – a phenotype associated with reduced proliferation.

We agree that it is widely accepted that most of the well-investigated microcephaly conditions were shown to be associated with an abnormal size of the of progenitor's pool resulting from altered proliferation, differentiation and/or cell survival. Indeed, mutated genes associated to microcephaly (ASPM, Microcephalin, CDK5RAP2, CENPJ, etc.)(1, 2) often play a role in cell division during neurogenesis. This cellular mechanism represents, of course, a relevant hypothesis for gamma tubulin-related disorders as TUBG1 protein is known to be essential for mitotic spindle formation during mitosis (3–6). However (and to our surprise), we show in this study that proliferative processes seem not to be affected in TUBG1-associated microcephaly. In order to explain this apparent discrepancy and address the comment pointed out by the referee, we also studied whether TUBG1 disease-causing variants might not lead to primary microcephaly but rather secondary microcephaly, that mainly manifests in

postnatal stages. To address the referee's comment and provide evidences in favor of this hypothesis, we retrieved -whenever possible- the head circumference at birth from patient's medical files and evaluated knock-in brain size at P0. Interestingly, in patients for whom HC size at birth was available, values appeared within the normal range: p.L387P, HC=31cm; p.Y92C, HC= 33cm, but then progressed to severe microcephaly p.L387P, HC= -5,5 SD; p.Y92C, HC= -4SD. In the knock-in mouse model, our additional investigations also demonstrate that there are no differences in global brain size nor for any specific region at birth. To take into account the referee's comment, we included the previous paragraph in the discussion of revised version of the manuscript (lines 406-410). Supporting findings are included in Suppl. figure 9a, showing the absence of microcephaly in the KI model at P0.

Regarding the interesting question of why mutations in a native centrosomal protein, known for its important role in mitosis, seems to be only affecting post-mitotic processes, this has been extensively discussed in the manuscript in lines 417-461. In continuation with a paragraph highlighting results related to this question we included the following sentences: "...Our data suggests that disrupted neuronal migration could be related to additional non-centrosomal functions of γ -tubulin. In line with this hypothesis, centrosome independent roles for γ -tubulin in the control of microtubule dynamics has been reported in *D. melanogaster* (7). Authors show that γ TuRCs localize along interphase microtubules and suggest they stabilize them by limiting catastrophe events at the plus tips. Similarly, in *S. pombe*, γ -tubulin has been suggested to cooperate with the kinesin-like protein Pkl1p to regulate the dynamics and organization of microtubules(8). In HeLa cells, specific inhibition of γ -tubulin with gatastatin decreased the dynamics of interphase microtubules without affecting the overall microtubule network and the number of growing microtubules(9). More specifically, in post-mitotic neurons, γ TuRCs have been shown to participate in centrosome-independent microtubule nucleation mediated by Augmin complexes(10). Finally, during cortical development, it has been recently shown that acentrosomal microtubules nucleation controls neuronal morphology ensuring proper migration in vivo (11). Altogether, this supports our results showing that pathogenic gamma-tubulin leads to reduced microtubule dynamics in mutant cells during interphase, and a failure in neuronal migration".

Since gamma tubulin complexes are known to regulate microtubule dynamics, the claim that altered neuronal migration results from altered microtubule dynamics, independently of centrosome integrity and function may not be entirely correct.

Though additional studies using electron microscopy (New Suppl. figure 5b) show that centrosome integrity is not affected, and we observe no MT nucleation defects (figure 4a), to avoid overstatements we removed this sentence from the revised version of the manuscript.

The authors should explain why Tubg1^{Y92C/+} was specifically selected for generating the mouse model over other mutations.

We would like to mention that there is no specific reason for this choice. It is only related to the chronology of mutation identification. A program aiming to develop several KI models (TUBG1 and other MCD-related genes) was initiated several years ago and TUBG1 Tyr92Cys mutation was chosen, as it was the first one our team identified and for which some preliminary data were generated. To clarify this point, we added in the revised version of the manuscript "...we generated a constitutive knock-in mouse model (Tubg1^{Y92C/+}) bearing the first identified human mutation Tyr92Cys" (line 255-256). However, as our interest is not limited to this specific mutation, a significant number of interpretations/assertions/conclusions have been made with all four mutations using in utero electroporation and in cellulo experiments (positioning, proliferation, migration, specification, microtubules dynamics, gamma tubulin complexes formation), suggesting that results obtained by the knock-in model could mostly be extended to other pathogenic variants.

Suggestions

1) Results obtained with experiments where a given protein is overexpressed should be interpreted with caution. This is because even a wild type protein expressed above its normal physiological expression level can potentially produce a phenotype due to toxic gain of function (e.g. Figure 4 E). Therefore, Tubg1 RNAi rescue by expression of RNAi resistant wild type or mutant Tubg1 should be performed at least in a small number of experiments.

We of course agree with the referee and to address this recommendation, we have now included this experiment in the new version of the manuscript (lines 138-141 and in the Suppl. figure 2d-e). We show that TUBG1 rescues the misposition phenotype observed in TUBG1 downregulated cells whereas the phenotype is potentiated when expressing mutant TUBG1. We also show that the overexpression of WT TUBG1 does not affect the localization of neurons within the cortical plate (Suppl. Figure 2c) Additionally, in all experiments where there is an overexpression of TUBG1 variants, statistical comparisons have been made between the control empty vector and the overexpression of the wild-type protein. This can be found in figures, and statistical details are included in Supplementary Table 3.

Also, the results of variant overexpression should be compared to those of wild type Tubg1 overexpression (as opposed to empty vector expression) with statistical tests, where possible.

As requested by reviewer in the new version of the manuscript, we have now compared the effect of the mutations with the overexpression of the wild type form of TUBG1. This is now included in the revised version of figures, and all the statistics have been detailed in supplementary Table 4. Whenever an effect of the overexpression of the wild type form of TUBG1 was observed, we have also mentioned it within the text, and in this case, mutants have been compared with wild type overexpression (see lines 233-234).

2) Why are different promoters used to drive the expression of a specific variant (DCX) and the reporter (NeuroD) in post mitotic neurons? This is also the case with experiments in radial glial progenitor cells (CAG promoter for driving the expression of a variant and BLBP for the reporter). *Though DCX is a postmitotic neuronal marker, the overexpression of TUBG1 with this promoter is not really high. This makes it a really good option, especially in our case where we are working with heterozygous missense mutations, to avoid high overexpression and to be as close as possible to the disease conditions. However, it leads to an equally low GFP expression level, resulting in a weak staining and making it harder to work with these images, specially postnatally. As NeuroD is a much stronger promoter that is expressed in the same cell population, we co-electroporated this construct to have the best combination of low TUBG1 overexpression along with high GFP overexpression in the same cells. Regarding the co-electroporation with BLBP-GFP and CAG-driven TUBG1 variants, we thought it is a reasonable strategy for the following reason: CAG promoter is an ubiquitously expressed promoter which therefore, leads to the expression of the different TUBG1 variants starting from progenitor state up to differentiated neurons, whereas BLBP is a radial glia specific marker known for their role as precursors and as a support for migrating neurons (12, 13). In this context, our question was whether expression of TUBG1 variants in progenitors and post-mitotic neuronal cells (the reason for CAG-driven TUBG1 expression) is associated with a phenotype in radial glia fibers organization that can be visualized with the BLBP-GFP construct, making the combination of these two promoters, a reasonable choice for the observation we needed.*

3) Time-lapse images or videos for should be provided for live imaging of neuronal migration since these were used for analyzing the velocity, number of pauses, pause duration in migrating neurons. Similarly, images of EB3 comets should be provided along with representative kymographs. *These videos have now been added as Supplementary data.*

4) The authors claim to address the functional redundancy of Tubg1/2 by showing a partial rescue of phenotype produced by mutant form of Tubg1 by overexpression of TUBG2. However this is experiment does not address the redundancy between the two genes. It would be addressed by the rescue of Tubg1 knockdown phenotype by Tubg2 overexpression or vice versa.

We thank reviewer for pointing out this interesting fact. As suggested, and to better characterize a possible redundancy between Tubg1 and Tubg2, we have studied whether Tubg2 overexpression was capable of restoring Tubg1 knock-down and TUBG1 disease causing mutations phenotypes (lines 134-141; lines143-146 and new suppl. figure 2d-e). Altogether, these additional results along with rescue experiments using TUBG1 mutants (Suppl. figure 2f-i), actually show that TUBG2 can only partially recover the phenotype observed by Tubg1 down-regulation and mutant's overexpression. This argues in favor of a limited functional overlap between these two isoforms (lines 146-148).

5) Expression of the mutant forms of Tubg1 leads to the arrest of neurons in the intermediate zone. The arrested neurons show multipolar morphology. However, the defect in the multipolar to bipolar transition should be quantified.

Arrested neurons in the IZ present in fact bipolar morphology. To address referee's comment, we have now assessed the different morphological transitions of neurons in time-lapse experiments where TUBG1 constructs are overexpressed. This data is now included in additional panels f and g of figure 3 and show that there are no differences in these parameters. Additionally, we have broadened our study, and performed the same type of analysis in time-lapse recordings using Tubg1^{Y92C/+} mice. Results obtained using this model, where wild type and mutant TUBG1 proteins are expressed in physiological levels, confirm the lack of effect of the mutation in morphological transitions at this stage (please see figure 5k,l).

Reviewer #2 (Remarks to the Author):

In this manuscript the authors explore the cellular and molecular defects that are associated with TUBG1 mutations and cortical malformations. They have employed a wide range of sophisticated methods including: in utero electroporation; live cell time lapse imaging of migrating neurons; cellular analysis of microtubule dynamics; the generation of in vivo mouse models; and EEG analysis. They have made a number of important findings - which focus on defects in post-mitotic neurons rather than the progenitors themselves. This (I think) is unexpected and interesting. In general the paper is well written, easy to follow, and makes a valuable contribution to the field.

I think this paper merits publication in Nature communications, but first requires some work. While the manuscript has numerous strengths in its current form there are number of aspects that I found deeply frustrating. First, the authors have consistently failed to specify the n numbers in the figure legends nor have they used dot plots. Accordingly, its very difficult for the reader to assess whether their data really does justify their conclusions. Secondly, their mouse model has been analyzed superficially, despite being the punch line of the abstract. Finally, there are a number of experiments which require more information with respect to the controls used and how they were quantitated.

Accordingly, I would suggest the following improvements.

Major Issues:

1. The authors should specify in the main text what control was used for their fibroblast experiments. The control used is critical to the interpretation of their patient data. Were the controls parental fibroblasts? This would be ideal. A single control from an unrelated individual is not sufficient to draw any conclusions (3 unrelated control individuals is a minimum).

We agree with reviewer's suggestion. Unfortunately, we do not have access to parental fibroblasts and therefore have used unrelated control fibroblasts. For this reason, we included two additional controls to the experiments performed with fibroblasts. This point has been specified in the main text (lines 224-225) and the new data that includes the two additional control fibroblasts are included in figure 4a-c, f, g.

Moreover, the authors should quantitate the "shorter and less organized" microtubules following depolymerisation, rather than relying on qualitative assertions.

This quantification has been added to figure 4b, c.

2. Further validation of their mouse model is required. Specifically, the authors should performed western blot analysis at an appropriate developmental time point to assess the levels of TUBG1 protein in their Y92C/+ animals, as well as the knockout animals. Is there a truncated protein in their KO animals? They should likewise assess the levels of mRNA in these animals.... is there nonsense mediated decay in their KO animals? Whereabouts is the termination mutation in the KO animals?

Results illustrating requested control experiments (expression studies by Western Blots, RT-PCR and sequencing) are now included in the supplementary figure 6.

- *WB using brain protein extract during development (embryonic day E12, E14, E18 and P3) showing similar protein levels in the Tubg1^{Y92C/+} and WT samples.*
- *Sequencing results of RT-PCR products using brain total RNA extracted from control and Knock-in models and primers located in exons 1 and 4 showing the presence in the KI model of the variant at heterozygous state (with similar height of the peaks corresponding to the wild and mutant alleles).*
- *Quantitative RT-PCR using brain total RNA extracted from control and Knock-in models that did not reveal any consistent expression difference of Tubg1 transcripts.*
- *For the Knock-out model, deletion of the LoxP flanked exons 2 and 3 of Tubg1 mRNA: NM_134024.3, c.50_330del is expected to lead to a disruption of the reading frame (frameshift), p.Ile17Argfs*6. As illustrated in Suppl. figure 6, WB and quantitative RT-PCR show reduced levels of TUBG1 protein and transcript, Also, RT-sequencing of RT-PCR products using brain total RNA of the heterozygous KO model and primers located in exon 1 and 4 shows the presence of a normal transcript and a transcript lacking exons 2 and 3. It is worth mentioning that this latter transcript contains also an additional « aberrant » sequence corresponding to a short intronic sequence (intron 1 of Tubg1), the remaining LoxP (as a result of the recombination) and a short sequence of intron 3. This additional sequence is flanked by splicing sites located in intron 1 and 3. The likely explanation is that this aberrant transcription lacking exons 2 and 3 results from an activation of the cryptic splice sites following the deletion caused by the Cre recombinase.*

3. The analysis of their mouse mutants is superficial and currently under utilizes this valuable resource. The authors have not analyzed the cortex in sufficient depth for their Y92C/+ animals or their KO animals. An analysis of lamination in adult animals should be performed (e.g. Foxp2, ER81, Cux1), and cortical thickness assessed. Is the number of interneurons similar?

In order to take into account referee's comment, we performed a series of experiments that allowed us to more thoroughly characterize our mouse model.

First, we studied radial glial cells proliferation as well as the distribution of the different progenitors' pools at several developmental stages (E12, E14 and E16) (Suppl. figure 7). We have also included a more detailed analysis of lamination and cortical thickness using classical layer markers at late stages of development (figure 5a,b). In terms of the mechanisms affected by the mutation, we performed time-lapse studies of migrating neuron, where we assessed different parameters of migration dynamics, we studied the morphological changes that occur in neurons and that precede migration, and the localization of the centrosome during this phase (figure 5f-m).

As suggested by referee, we also refined our morphological analyses of Tubg1^{Y92C/+} adult mice and included new results regarding lamination and cortical thickness. For this we relied on classical histological studies that have been proven to be useful for this purpose, and which resulted in new conclusive results: the presence of lamination defects in the cortex of adult mice (figure 6c). We also evaluated the timing of the microcephaly and the hippocampal phenotype observed in Tubg1^{Y92C/+} mice and performed both morphometric analysis at birth (Suppl. figure 9a) and histological studies at P2 (Suppl. figure 9b). These results show the absence of microcephaly at birth, suggesting a secondary microcephaly (related to postnatal brain development and growth defects that could result from altered neuronal maturation, synaptogenesis and connectivity), and the appearance of the hippocampal phenotype after E18 (please see also the response to the first comment of referee 1 highlighting available details regarding patients HC at birth).

Concerning interneurons, we also assessed whether the number and distribution of different types of interneurons was affected in Tubg1^{Y92C/+} adult mice (calbindin+ and calretinin+). These results have been added in Suppl. figure 8 and a sentence saying "no defects in interneurons number nor distribution were found in Tubg1^{Y92C/+} mice" in lines 316 of the main text.

4. The authors should assess neurogenesis in their mouse mutants, reflecting the experiments they have performed in utero. In vivo models are always better, and it would be valuable to confirm that neurogenic output is similar in their mice. An assessment of PH3 at E14.5 for instance would be valuable, as well as the number of SOX2 positive progenitors.

As recommended by the referee, these experiments have been performed and added in the manuscript (lines 280-284) and associated to new Suppl. figure 7.

Stating in the discussion "data now shown" is not good enough. The words "data not shown" should not appear in any scientific manuscript.

We thank reviewer for this suggestion. To ameliorate the quality of our manuscript and to take into account referee's comment the words "data not shown" have been removed from the manuscript. All the data showing the lack of neuroanatomical and behavioral abnormalities in Tubg1^{+/-} mice are now shown and have been added to new Suppl. figure 11. We have also produced new Suppl figure 9b where we depict the lack of ectopic neurons in the hippocampus at late embryonic stages (E18); as well as we added a new panel to figure 5, showing that ectopic cortical neurons are SATB2-positive.

5. The methods should include a section that describe how sections were quantitated.

Moreover, all quantification should be performed blinded to genotype or condition. If this has not been done, the data should be re-analyzed.

We have added this information in the Materials and Methods sections, where we specify how experiment were quantified and that analysis were done blinded to genotype.

6. I would restructure the presentation of their mouse data - beginning with the overall volumetric assessment of the brain regions. From there you could then focus on the cortex, hippocampus and cerebellum in a sequential manner. Building up Figure 5, and leaving Figure 6 just for behavior and EEG.

We have taken this comment into account and re-structured the manuscript as suggested. In the new version of the article neuroanatomical and morphometric studies are included in figure 6, whereas behavioral analysis and EEG can be found in figure 7.

7. Figure 6 presents the results of their behavioral experiments. I have some concerns about this. Were sex matched littermate controls used for their experiments? It is strange that 19wt animals and 17 TUBG1 were used. Were all experiments performed blind to genotype? Moreover, if indeed the ages ranged from 2.5 to 6.5 months of age this is a major concern. Only data from animals

of the same age should be shown (+/- a week). If there is a degenerative aspect to this phenotype - using animals of different ages is problematic.

We agree with referee's comment regarding the cohorts used for behavioral experiments. Animals were sex matched (male) and experiments have been performed blinded to genotype as detailed in the materials and methods section. To get a first glimpse about behavioral status of the TUBG1 Knock-in model, we performed a first pilot study using available animals (10 WT and 8 Tubg1^{Y92C/+} mice) with ages ranging from 2.5 to 6.5 months that allowed to point out behavioral defects. To further strengthen these results, we performed a second study using 9 WT and 9 TUBG1^{Y92C/+} mice and presented the data generated for the two cohorts. To address referee's comment and avoid the justified criticism regarding the wide range of ages of animals used in the study, we included in the revised version of the manuscript, results obtained only from the second cohort of 9 WT and 9 TUBG1^{Y92C/+} animals with ages ranging from 10 to 12 weeks. The major conclusions by using this cohort, do not vary from those described in the previous version of the manuscript. Only slight modifications in terms of statistics were found but without changing the fundamental message. All of these new results are now included in figure 7 and Suppl. figure 10.

Review of Ivanova:

Minor issues:

1. The authors do not indicate the n number for the in utero electroporation experiments in either the figure legends, text or methods. You cannot expect your reader to download a supplementary table for this information. Please rectify this. How many sections were analyzed for each animal? Again this is not specified.

As suggested by reviewer, we have now added n numbers in all figure legends to make reading smoother. For all in utero electroporation experiments, a minimum of 3 different sections were analyzed. This information has been included in the materials and methods section.

2. The Figure legends for supplementary figure 1, do not actually match supplementary figure 1. Should the first panel be deleted?

We thank reviewer for pointing out this mistake. Panel A of figure S1 has been removed. Now the figure matches the figure legend and they are correctly cited within the main text.

3. The n number for the PH3 experiments shown in Figure S2 is not specified. Again, this requires rectification. Moreover, all data should be presented as dot bar graphs so the reader can better appreciate the variation in the data set. An image showing the PH3 staining should also be included.

As suggested by reviewer, we have now added n numbers in all figure legends to make reading smoother. We have also modified, whenever possible, the figure style by replacing our bar graphs by new dot bar graphs. Additionally, we have added representative images in figure S3 for both PH3 and Tbr2/Pax6 stainings.

4. The authors state on page 9 lines 272 that "Our results indicate that disrupted locomotion in neurons expressing recombinant TUBG1 is not due to defects in centrosome positioning". This statement requires some clarification, as it might be interpreted as WT recombinant TUBG1..... and given the centrosomal phenotype of those mice expressing TUBG1-Tyr92Cys this assertion is confusing.

*We thank reviewer for referring to this possibly confusing sentence in the manuscript. To clarify this point, we modified this phrase to "disrupted locomotion in neurons expressing **pathogenic** recombinant TUBG1".*

5. The n numbers for comets and cells analyzed for the microtubule dynamics experiments should again be specified in the figure legends for Figure 4, and again dot and bar graphs employed. How many different times was this experiment replicated? Were they independent transfections?

As suggested by reviewer, we have now added n numbers in all figure legends to make reading smoother. In the case of EB3 comets experiments, and as detailed in the new figure legend, this means three independent experiments (different cell cultures and different transfections). In all cases (transfected cells and Fibroblasts experiments) in each experiment and for each condition 10 cells were analyzed, and for each cell we quantified between 15-30 comets. This last information is now detailed in the materials and methods section.

6. On panel H of Figure 4, the TUBG1-Leu387Pro pull-down results in a noticeably lower amount of Gamma tubulin.... while the TAG-RFP seems similar to the other constructs. How do the authors explain this?

Densitometric analysis of the gels mentioned by reviewer, indeed show that there is a decrease in the amount of gamma-tubulin that is pulled-down with tag-RFP. However, this is the case for all mutants when compared to the immunoprecipitation of the RFP-WT TUBG1 (please see Suppl. figure 5c). This was also detailed in the main text (lines 246-247). This reduction suggests that all mutant TUBG1 proteins have a decrease ability to associate with endogenous wild type gamma-tubulin. Therefore, although the degree of change might be different among different mutants, the overall effect is the same: a reduction in the amount of immunoprecipitated gamma-tubulin.

7. With respect to Figure 4, it would be useful to show the full western blot gels with a ladder in the supplementary data.

We have now prepared new Supplementary figure 12 which contains full images of all our western blots with the molecular weight marker.

8. Please expand supplementary Figure 5, showing the southern blot results demonstrating correct homologous recombination in ES cells. How were animals subsequently genotyped?

As suggested by referee, southern blot showing the correct homologous recombination in ES cells has been added as a new panel b in Suppl. Fig.6

Concerning the genotyping strategy used, a new paragraph has been included in the materials and Methods section (lines 648-658), and the position of the primers used are now represented in Suppl. figure 6a.

9. What Cre line was employed to generated KO animals?

The generation of the heterozygous knock-out mice was done using the following Cre line: Highly-efficient, fluorescent, locus directed cre and FlpO deleter mice on a pure C57BL/6N genetic background.

Birling MC, Dierich A, Jacquot S, Hérault Y, Pavlovic G. Genesis. 2012 Jun;50(6):482-9. doi: 10.1002/dvg.20826. Epub 2012 Mar 20. PMID:22121025.

This information can be found in the materials and methods section of the manuscript (line 647).

10. The TBR1 staining shown in Figure 5A is not informative. Layer VI? Everything seems to be stained.

We agree with reviewer in this observation. In the new version of the manuscript, this figure has been modified and additional analysis have been made.

11. Again no n numbers shown in Figure legend 6.

As suggested by reviewer, we have now added n numbers in all figure legends to make reading smoother.

Reviewer #3 (Remarks to the Author):

It has known that TUBG1 variants in patients link to cortical dysgeneses associated with intellectual disability and epilepsy. However, the underlying mechanisms remain elusive. In this manuscript, Ivanova and colleagues investigate four human MCDs-related TUBG1 missense variants on cortical development by using in-utero electroporation and a knock-in Tubg1(Y92C/+) mouse model. They found that TUBG1 variants affect neuronal migration but do not effect on progenitor proliferation. Using time-lapse videomicroscopy, authors further demonstrated TUBG1 variants affect microtubule dynamics rather that centrosomal nucleation ability. In addition, they reported that Tubg1(Y92C/+) mouse showed neuroanatomical and behavioral defects.

Although the manuscript clearly represents a significant amount of work and contains interesting data, I have some suggestions/concerns.

We thank reviewer for his comments and constructive recommendations. As suggested, additional work has been performed and the manuscript has been modified in order to address referee's questions.

1. To highlight the significant of this manuscript, it's necessary to further emphasize these four TUBG1 variants are human MCDs-related variants, and cite relative literatures.

To acknowledge reviewer's recommendation, we have stressed this point throughout the manuscript : the abstract opens with the statement saying that TUBG1 variants are linked to human cortical malformations. In line with this, we dedicate the fifth paragraph of the introduction, to the description of all the (to our best knowledge) reported variants associated to this kind of pathology and where we cite the related articles. Also, additional changes regarding this point have been made and highlighted within the revised version of the manuscript.

2. Although the typical image data (Fig. S1D) show that GFP+ cells electroporated TUBG1 WT and four TUBG1 variant vectors overexpress TUBG1, authors should further quantify the percentage of overexpressed GFP+ cells in GFP+ cells in each group.

We have now added in panel a and b of Suppl. figure 1 images showing the full electroporated cortical area of the different TUBG1 variants that were also stained against TUBG1, showing the overexpression

of TUBG1 in the majority if not all GFP+ cells. Quantifications (Suppl. figure 1c) show that a similar percentage of GFP+ cells express either forms of TUBG1.

3. Please explain why quite number of Cux1+/ RFP- cells in four TUBG1 variants locate in the lower layer of neocortex (Fig. 1D and 1E).

We do agree with this observation. As we suspected this was due to a loss of GFP signal that occurs with time. We performed new electroporations with a reporter that has a brighter signal and is maintained for longer periods of time (mVenus)(14). In these new experiments we could observe that the majority of ectopic Cux-1 positive cells at E18.5 were positive for the transfection reporter (new figure 1d). This was also the case in P8 and P21 mouse brains electroporated at E14, for the upper layer markers Cux-1 and Satb2, as can be observed in new figure 1e and Suppl. figure 4.

Moreover, to exclude the possibility that mislocalized cells co-express lower layer markers, Ctip2 and Tbr1 markers should be tested in these cells. Similarly, in DCX promoter experiments (Fig. 2), co-expression of Cux1, Ctip2 and Tbr1 markers with electroporated neurons should be examined.

We have included in this new version of the manuscript stainings showing that mislocalized electroporated neurons express only upper layers marker (Cux1, Satb2) and not lower layers markers (Ctip2 and Tbr1). This has been done for electroporations performed with both DCX and CAG promoters, and assessed at embryonic, early post-natal and relatively mature post-natal stages (E18, P8, P20). All of this new data can be found in figure 1d,e; Suppl. figure 1e (for CAG experiments) and in Suppl. figure 4 b-f (for DCX experiments). This new information has been also detailed in the main text related to these figures.

4. Through the quantifications (Fig. 1C and Fig. 2B), we can observe the phenotype of CAG promoter is much severe than that of DCX promoter. Please discuss possible reasons.

We thank reviewer for highlighting this very interesting observation. The CAG promoter is a very strong ubiquitous promoter which therefore, leads to a very high expression of the different TUBG1 variants starting from progenitor state up to differentiated neurons. On the other hand, DCX driven expression is very much subtle and specific to post-mitotic cells (15–18). Therefore, using this latter is a common strategy used to study the effects of a certain protein specifically in post mitotic neurons, deleting the possible effect on proliferative progenitors. Different expression levels driven by these two promoters

can be clearly observed by comparing the TUBG1 protein levels between Suppl. figure 1a,b (CAG driven TUBG1 expression in magenta) and Suppl. figure 4a (DCX driven TUBG1 expression)

Given that TUBG1 mutations do not affect any proliferative process when analyzed by the CAG driven expression of different mutants, we assume that the difference in the severity of the phenotype while using these two different constructs, comes from the differential levels of expression achieved by using each promoter. This is in line with what we observe in the mouse model. Here again the heterozygous expression of the Tyr92Cys mutation does not lead to proliferation alterations nor any unbalance in apical and intermediate progenitors as can be observed in new Suppl. Figure 7a-c. However, we do have layering problems and neuronal migration is defective. These observations are much less severe than those observed with both CAG and/or DCX driven overexpression, probably due to the lower amounts of mutant protein expressed.

5. To test the neuronal mispositioning is due to an arrest rather than a delay in neuronal migration, authors quantified the distribution of electroporated neurons in postnatal stages, CAG promoter at P8 and DCX promoter at P3. However, as compare with neuronal distribution at E18 in DCX promoter experiments (Fig. S3B), electroporated neurons can still migrate at P3 (Fig. 2B). Therefore, I suggest to quantify the distribution of electroporated neurons for both CAG and DCX promoter experiments at relative mature postnatal stage (e.g., P20).

We understand reviewer's concern. In order to be more convincing about the arrest of mutant TUBG1-expressing neurons, we have performed additional electroporation experiments and analysis at P20. We now show that in both scenarios (CAG and DCX promoters) neurons remain arrested and express upper layer markers even 20 days after birth, leaving aside any potential postnatal migration and recovery of the phenotype. Please refer to new figures 1e, Suppl. figures 1e and Suppl. figure 4 c-f.

6. In rescue experiments, co-electroporation of TUBG1 or TUBG2 can partially rescue Tyr92Cys TUBG1 variant-induced positioning phenotype. It will be very helpful to test whether co-electroporation of TUBG1 or TUBG2 can rescue other TUBG1 variant-induced phenotypes.

We have now performed these experiments and included them in Suppl. figure 2h-i. These experiments show that regardless the TUBG1 mutation, the effect of co-expression of TUBG1 is a complete reversion of the phenotype, whereas TUBG2 is able to at least partially rescue the phenotype.

In addition, authors mentioned that “Co-electroporation of either isoforms leads to a rescue in the neuronal positioning phenotype in a dose-dependent fashion”. Although it is correct for TUBG2, the rescue in the neuronal positioning phenotype for TUBG1 seems not to be in a dose-dependent fashion, since there is no significant change for neuronal positioning between two TUBG1 electroporated concentrations (Fig. S1H).

We thank reviewer for highlighting this point and we apologize for this inaccuracy. In the revised version of the manuscript, we now say “Co-electroporation of either isoforms leads to a rescue in the neuronal positioning defect, nonetheless, while a significant restoration of the phenotype was obtained with TUBG1, rescue with TUBG2 was only partial and dose-dependent. However, TUBG2 rescue was significant for other mutants.” (lines 143-146). In line with this modification, we have also substituted the representative image for Tyr92Cys + TUBG1 so that it reflects the quantifications and statistics.

7. Though current data, it's really hard to conclude “variants affect neuronal migration dynamics but not centrosomal positioning”. In other words, there is no direct evidence to support Tyr92Cys variant do not affect centrosomal positioning and further influence the neuronal migration. In order to demonstrate that Tyr92Cys variant still influences the neuronal migration even under centrosome migrating position, authors should further compared the percentages of migrating neurons and non-migrating neurons whose centrosome was positioned ahead of the nucleus between control-empty vector and DCX-TUBG1-Thr331Pro groups.

*We understand this concern and we agree that this is an important issue. To take into account referee's comment we decided to perform additional experiments to clarify this issue. First, keeping in mind that we have an accumulation of arrested bipolar neurons for both mutants, we analyzed whether there were any defects in morphological transition ruling out the influence of these transitions in bipolar neurons accumulation (new figure 3f, g). Then, as we see an increased proportion of arrested mutant neurons with centrosomes ahead of the nucleus, we better characterized what population of arrested neurons was affected by this reorganization. We observed over 50% increase of arrested bipolar neurons with the centrosome positioned ahead of the nucleus in mutant neurons, suggesting that neurons adopt the correct morphology for migration, achieve the correct positioning of the centrosome but still fail to initiate the migration (new figure 3k). Importantly, all of these data were further validated by time-lapse experiments performed in organotypic cultures of the *Tubg1*^{Y92C/+} mice, additionally supporting this idea in a physiological condition devoided of overexpression effect (new figure 5f-m).*

8. For Tubg1(Y92C/+) mouse, the phenotypes are much weaker than CAG and DCX promoter electroporation experiments. I suggest to systematically quantify the distribution of cortical layers in Tubg1(Y92C/+) mouse at relative mature postnatal stage (e.g., at P20).

As suggested by referee, we refined our morphological analyses of Tubg1^{Y92C/+} adult mice and included new results regarding lamination and cortical thickness. For this we relied on classical histological studies that are as useful for this purpose, which resulted in new conclusive results (figure 6c).

Moreover, it's worthy to further examine the neuronal migration and centrosome positioning in Tubg1(Y92C/+) mouse at embryonic stage through EGFP vector co-electroporation.

We have taken this recommendation into account. This experiment, along with multiple analysis regarding dynamics of migrating neurons, morphological changes preceding migration and centrosomal positioning has been added to the manuscript in the "Knock-in Tubg1^{Y92C/+} embryos show abnormal cortical layering" result section and are included as panels f-m of figure 5.

9. Authors should discuss why Leu387Pro variant does not affect the microtubule dynamics, but severely influence the neuronal positioning.

Indeed, the Leu387Pro variant of TUBG1 showed no alterations in microtubule's dynamics by using this specific strategy. However, whether this technique is sensitive enough to detect slight modifications on MT dynamics is not clear. However, though it is not statistically significant, there is nevertheless a decrease in the mean velocity of EB3 comets. To take into account referee's comment we added the following comment in the discussion "Additionally, our EB3 studies showed reduced microtubule dynamics for three of the recombinant forms and a slight decrease that did not reach statistical significance for the Leu387Pro variant, probably due to a lack of sensitivity of the technique." (lines 433-435).

10. It is nice to find mutant Tubg1(Y92C/+) mice show an increased epileptic cortical activity. However, the more detail information should be included. For example, how many Tubg1(Y92C/+) mice showed epileptic cortical activity.

*We appreciate this remark. We have now clarified this point by saying “Spontaneous cortical spike-waves were observed in **all three** *Tubg1*^{Y92C/+} mice....” In lines 347, and “ ... tonicoconvulsive seizure was recorded in **all analysed** *Tubg1*^{Y92C/+} mice but not in WT.” in line 353.*

Additionally, in order to facilitate article reading, we have complemented Supplementary table 3 (which contains all statistical parameters of all analyzed data in detail) with the n number for each experiment (in this particular case the number of animals used) in each respective figure legend.

Minor Comments:

1. There are too many “data not shown” in this manuscript. At least, authors should provide the original data that *Tubg1*^{+/-} mice showed no major abnormalities in their neuroanatomy and no behavioral alterations.

*We thank reviewer for this suggestion. To ameliorate the quality of our manuscript and to take into account referee’s comment the words “data not shown” have been removed from the manuscript. All the data showing the lack of neuroanatomical and behavioral abnormalities in *Tubg1*^{+/-} mice are now shown and have been added to new Suppl. figure 11. We have also produced new Suppl figure 9b where we depict the lack of ectopic neurons in the hippocampus at late embryonic stages (E18); as well as we added a new panel to figure 5, showing that ectopic cortical neurons are SATB2-positive.*

2. Authors should provide the image data that stained PH3, Pax6, Tbr2, and EGFP/RFP.

New representative images depicting Pax6/Tbr2 and PH3 staining in E16.5 in utero electroporated brain sections have been added to Suppl. figure 3.

3. Fig. S1A is redundant, since it has been already shown in Fig. 1A. Moreover, the number of the supplementary Figure 1 marked in alphabetical order in main text and figure legend is incorrect. Please revise.

We thank reviewer for pointing out this mistake. Panel A of figure S1 has been removed. Now the figure matches the figure legend and is correctly cited within the main text.

4. Authors electroporated pCAGGS-IRES-Tomato reporter, but showed GFP in Fig. S1D (should be Fig. S1C). Please check this.

This error has been corrected.

5. In line 394, “ μ -aminobutyric acid” should be “ γ -aminobutyric acid”.

This mistake has now been rectified.

Bibliography

1. Fernández,V., Llinares-Benadero,C. and Borrell,V. (2016) Cerebral cortex expansion and folding: what have we learned? *EMBO J.*, **35**, 1021–44.
2. Pang,T., Atefy,R. and Sheen,V. (2008) Malformations of cortical development. *Neurologist*, **14**, 181–91.
3. Oakley,B.R., Oakley,C.E., Yoon,Y. and Jung,M.K. (1990) Gamma-tubulin is a component of the spindle pole body that is essential for microtubule function in *Aspergillus nidulans*. *Cell*, **61**, 1289–301.
4. Stearns,T., Evans,L. and Kirschner,M. (1991) Gamma-tubulin is a highly conserved component of the centrosome. *Cell*, **65**, 825–36.
5. Joshi,H.C., Palacios,M.J., McNamara,L. and Cleveland,D.W. (1992) γ -Tubulin is a centrosomal protein required for cell cycle-dependent microtubule nucleation. *Nature*, **356**, 80–83.
6. Wiese,C. and Zheng,Y. (2000) A new function for the gamma-tubulin ring complex as a microtubule minus-end cap. *Nat. Cell Biol.*, **2**, 358–64.
7. Bouissou,A., Vérollet,C., Sousa,A., Sampaio,P., Wright,M., Sunkel,C.E., Merdes,A. and Raynaud-Messina,B. (2009) γ -Tubulin ring complexes regulate microtubule plus end dynamics. *J. Cell Biol.*, **187**, 327–334.
8. Paluh,J.L., Nogales,E., Oakley,B.R., McDonald,K., Pidoux, a L. and Cande,W.Z. (2000) A mutation in gamma-tubulin alters microtubule dynamics and organization and is synthetically lethal with the kinesin-like protein *pk1p*. *Mol. Biol. Cell*, **11**, 1225–39.
9. Chinen,T., Liu,P., Shioda,S., Pagel,J., Cerikan,B., Lin,T.C., Gruss,O., Hayashi,Y., Takeno,H., Shima,T., et al. (2015) The γ -tubulin-specific inhibitor gatastatin reveals temporal requirements of microtubule nucleation during the cell cycle. *Nat. Commun.*, **6**, 1–11.
10. Sánchez-Huertas,C., Freixo,F., Viais,R., Lacasa,C., Soriano,E. and Lüders,J. (2016) Non-centrosomal nucleation mediated by augmin organizes microtubules in post-mitotic neurons and controls axonal microtubule polarity. *Nat. Commun.*, **7**, 12187.
11. Cunha-Ferreira,I., Chazeau,A., Buijs,R.R., Stucchi,R., Will,L., Pan,X., Adolfs,Y., van der Meer,C., Wolthuis,J.C., Kahn,O.I., et al. (2018) The HAUS Complex Is a Key Regulator of Non-centrosomal Microtubule Organization during Neuronal Development. *Cell Rep.*, **24**, 791–800.
12. Feng,L., Hatten,M.E. and Heintz,N. (1994) Brain lipid-binding protein (BLBP): A novel signaling system in the developing mammalian CNS. *Neuron*, **12**, 895–908.
13. Feng,L. and Heintz,N. (1995) Differentiating neurons activate transcription of the brain lipid-binding protein gene in radial glia through a novel regulatory element. *Development*, **121**, 1719–30.
14. Courchet,V., Roberts,A.J., Meyer-Dilhet,G., Del Carmine,P., Lewis,T.L., Polleux,F. and Courchet,J. (2018) Haploinsufficiency of autism spectrum disorder candidate gene *NUAK1* impairs cortical development and behavior in mice. *Nat. Commun.*, **9**, 4289.
15. Hand,R., Bortone,D., Mattar,P., Nguyen,L., Heng,J.I.-T., Guerrier,S., Boutt,E., Peters,E., Barnes,A.P., Parras,C., et al. (2005) Phosphorylation of Neurogenin2 Specifies the Migration Properties and the Dendritic Morphology of Pyramidal Neurons in the Neocortex. *Neuron*, **48**, 45–62.
16. Heng,J.I.-T., Nguyen,L., Castro,D.S., Zimmer,C., Wildner,H., Armant,O., Skowronska-Krawczyk,D., Bedogni,F., Matter,J.-M., Hevner,R., et al. (2008) Neurogenin 2 controls cortical neuron migration through regulation of *Rnd2*. *Nature*, **455**, 114–118.
17. Guerrier,S., Coutinho-Budd,J., Sassa,T., Gresset,A., Jordan,N.V., Chen,K., Jin,W.-L., Frost,A. and Polleux,F. (2009) The F-BAR domain of *srGAP2* induces membrane protrusions required for neuronal migration and morphogenesis. *Cell*, **138**, 990–1004.
18. Brown,J.P., Couillard-Després,S., Cooper-Kuhn,C.M., Winkler,J., Aigner,L. and Kuhn,H.G. (2003)

Transient expression of doublecortin during adult neurogenesis. J. Comp. Neurol., 467, 1–10.

Reviewers' Comments:

Reviewer #2:

Remarks to the Author:

This a revised manuscript from Hinckelmann and colleagues. In general the authors have gone to great lengths to address the comments of all three reviewers. Many new experiments have been added which support the conclusions of the authors. It is a considerable improvement, however, the authors have again failed to properly quantify some of these data, and still n numbers are lacking from numerous experiments. The authors should note that all three reviewers raised issues regarding the quantification of data. I had hoped to wave this paper through, but there are still issues that need to be addressed. To persuade your audience that your findings are real data should always be quantified and n numbers should always be specified.

Issues

- There are no error bars or n numbers shown for Supplementary Figure 6 d of f. Please remedy this.
- The levels of TUBG1 protein shown in Supplementary Figure 6e and Figure 6g, should again be quantified and the n number specified. These experiments should have a n=3 as a minimum.
- There are no n numbers specified for supplementary figure 9.
- There is no n number specified for supplementary figure 11 a,b, or c.
- Please specify the n numbers and statistics for supplementary figure 10 in the figure legend. It is not clear what the sentence "All mice performed at the same level" means.
- The new electronmicrographs shown in Supplementary figure 5b are not clear. The protofilament structure is not visible, nor is the n number specified. How many centrioles were analyzed? The authors might consider adopting a more cautious tone on this front. They now state in the abstract that there are no "major structural nor functional centrosome defects".... But they haven't actually looked at this in their mouse model.
- The authors show that overexpression of TUBG2 fails to rescue a TUBG1 knockdown with a sh-RNA. The sequence of this sh-RNA should be specified (it is not in the original Porier manuscript), and can the authors confirm that it does not target TUBG2.

Reviewer #3:

Remarks to the Author:

The authors have addressed my concerns and comments satisfactorily.

REVIEWERS' COMMENTS:

Reviewer #2 (Remarks to the Author):

This a revised manuscript from Hinckelmann and colleagues. In general the authors have gone to great lengths to address the comments of all three reviewers. Many new experiments have been added which support the conclusions of the authors. It is a considerable improvement, however, the authors have again failed to properly quantify some of these data, and still n numbers are lacking from numerous experiments. The authors should note that all three reviewers raised issues regarding the quantification of data. I had hoped to wave this paper through, but there are still issues that need to be addressed. To persuade your audience that your findings are real data should always be quantified and n numbers should always be specified.

We thank reviewer for acknowledging the improvement in our manuscript, following extensive revisions of the first version. In this new round of revisions, we have particularly focused in making sure data are quantified, and that statistical parameters (n number, test used, p-value) are present in all figure legends, and that dot plots showing the distribution of single data are included in the figures whenever possible. However, though we agree with both the reviewer and editor that scientific data should normally be represented with appropriate statistics, when it comes to conveying a clear message from large datasets, we find that showing extensive tables or figures may become, in some cases, confusing. To avoid this drawback, we thought that extensive statistics can be provided in supplements (as we propose here) so that anyone can actually check the accuracy of the original figure we provided in detail.

Issues

- There are no error bars or n numbers shown for Supplementary Figure 6 d of f. Please remedy this.
This figure has now been re-formatted and error bars and distribution plots have been added
- The levels of TUBG1 protein shown in Supplementary Figure 6e and Figure 6g, should again be quantified and the n number specified. These experiments should have a n=3 as a minimum.
Figure 6e and Figure 6g have been modified, and they now include histograms showing densitometric quantification of the western blots. N numbers have been included in the figure legend
- There are no n numbers specified for supplementary figure 9.
The number of pups used in this experiment has been incorporated in the legend.
- There is no n number specified for supplementary figure 11 a,b, or c.
This information has been included in the corresponding legend
- Please specify the n numbers and statistics for supplementary figure 10 in the figure legend. It is not clear what the sentence "All mice performed at the same level" means.
In Supplementary Figure Legend 10 the n number, statistical test used, statistical significance, as well as definition of the box-plots parameters have been included.
- The new electronmicrographs shown in Supplementary figure 5b are not clear. The protofilament structure is not visible, nor is the n number specified. How many centrioles were analyzed? The authors might consider adopting a more cautious tone on this front. They now state in the abstract that there are no "major structural nor functional centrosome defects".... But they haven't actually looked at this in their mouse model.
In order to address reviewer's comment, we have included in the figure improved quality images of electronmicrographs showing perpendicular centrioles, and cross sections of centrioles showing the

nine triplets microtubules structure. The n number (4 independent experiments) and the number of cells analyzed (100 in average) has been included in the figure legend

Concerning the sentence reviewer pointed out, indeed, these studies have been performed in fibroblasts derived from subjects bearing the Tyr92Cys or Thr331Pro variants respectively and not in the mouse model, and to avoid an overstatement, we do say that there are no “major” structural nor functional centrosome defects. However, to be more cautious on this front, we also modified this sentence and now says “no major structural nor functional centrosome defects in subject-derived fibroblasts”

- The authors show that overexpression of TUBG2 fails to rescue a TUBG1 knockdown with a sh-RNA. The sequence of this sh-RNA should be specified (it is not in the original Porier manuscript), and can the authors confirm that it does not target TUBG2.

As requested, we have now incorporated the Tubg1 sh-RNA sequence in the Materials and Methods section (lines 487-489). To confirm the specificity of this sh-RNA, we performed efficiency studies and quantified the Tubg-1 and Tubg-2 mRNA levels in cell-extracts transfected with Tubg1 sh-RNA. As shown in the image below, sh-Tubg1 does not modify Tubg2 mRNA levels while it results in a significant decrease of about 50% for Tubg1 mRNA.

Tubg1 and Tubg2 relative mRNA levels detected by qPCR on cell extracts from cells transfected with control shRNA or sh-Tubg1. Data are normalized to actin and relative to mRNA levels in Control. Data represent mean +/- SEM, n=3, Students T-test.

Reviewer #3 (Remarks to the Author):

The authors have addressed my concerns and comments satisfactorily.